

# A general theory on frequency and time-frequency analysis of irregularly sampled time series based on projection methods. I. Frequency analysis

Guillaume Lenoir[1] and Michel Crucifix[1,2]

[1]Georges Lemaître Centre for Earth and Climate Research, Earth and Life Institute, Université catholique de Louvain, BE-1348, Louvain-la-Neuve, Belgium
[2]Belgian National Fund of Scientific Research, Rue d'Egmont, 5, BE-1000 Brussels, Belgium

*Correspondence to:* Guillaume Lenoir (guillaume.lenoir@hotmail.com)

**Abstract.** We develop a general framework for the frequency analysis of irregularly sampled time series. It is based on the Lomb-Scargle periodogram, but extended to algebraic operators accounting for the presence of a polynomial trend in the model for the data, in addition to a periodic component and a background noise. Special care is devoted to the correlation between the trend and the periodic component. This new periodogram is then cast into the

Welch overlapping segment averaging (WOSA) method in order to reduce its variance. We also design a test of significance for the WOSA periodogram, against the background noise. The model for the background noise is a stationary Gaussian continuous autoregressive-moving-average (CARMA) process, more general than the classical Gaussian white or red noise processes. CARMA parameters are estimated following a Bayesian framework. We provide algorithms computing the confidence levels for the WOSA periodogram that fully take into account the

uncertainty on the CARMA noise parameters. Alternatively, a theory using point estimates of CARMA parameters provides analytical confidence levels for the WOSA periodogram, which are more accurate than Markov chain Monte Carlo (MCMC) confidence levels and, below some threshold for the number of data points, less costly in computing time. We then estimate the amplitude of the periodic component with least squares methods, and derive an approximate proportionality between the squared amplitude and the periodogram. This proportionality leads to a

new extension for the periodogram: the weighted WOSA periodogram, that we recommend for most spectral analyses with irregularly sampled data. The estimated signal amplitude also permits filtering in a frequency band. Our results generalize and unify methods developed in the fields of geosciences, engineering, astronomy and astrophysics. They also constitute the starting point for an extension to the continuous wavelet transform developed in a companion article (Lenoir and Crucifix, 2017). All the methods presented in this paper are available to the reader in the Python

package WAVEPAL.



## 1   Introduction

In many areas of geophysics, one has to deal with irregularly sampled time series. However, most of state of the art tools for frequency analysis are designed to work with regularly sampled data. Classical methods include the discrete Fourier transform (DFT), jointly with the Welch overlapping segment averaging (WOSA) method, developed by (Welch, 1967), or the multitaper method, designed in (Thomson, 1982) and (Riedel and Sidorenko, 1995). Given the excellent results they provide, it is tempting to interpolate the data and simply apply these techniques. Unfortunately, interpolation may seriously affect the analysis with unpredictable consequences for the scientific interpretation (Mudelsee, 2010, p. 224).

In order to deal with non-interpolated, astronomical, data, (Lomb, 1976) and (Scargle, 1982) proposed what is now known as the Lomb-Scargle periodogram (denoted here LS periodogram). The LS periodogram is at the basis of many algorithms proposed in the literature, in particular, in astronomy, e.g. in (Mortier et al., 2015), (Vio et al., 2010), or (Zechmeister and Kürster, 2009), and in geophysics, e.g. in (Schulz and Stattegger, 1997; Schulz and Mudelsee, 2002), (Mudelsee et al., 2009), (Pardo Igúzquiza and Rodríguez Tovar, 2012), or (Rehfeld et al., 2011). More specifically, in climate and paleoclimate, the time series are often very noisy, exhibit a trend, and potentially carry a wide range of periodic components (e.g. see Fig. 6). Considering all these properties, we design in this work an operator for the frequency analysis generalizing the LS periodogram. The latter was built to analyze data which can be modeled as a periodic component plus noise. Since the periodic component may not necessarily oscillate around zero, (Ferraz-Mello, 1981) and (Heck et al., 1985) extended the LS periodogram, proposing an operator that is suitable to analyze data which can be modeled as a periodic component plus a constant trend plus noise. Their operator is designed to take into account the correlation between the constant trend and the periodic component, and is now a classic tool for analyzing astronomical irregularly spaced time series. In climate and paleoclimate, the periodic component may oscillate around a more complex trend than just a constant. This is why, in this work, we extend the previous result by proposing an operator that is suitable to analyze data which can be modeled as a periodic component plus a polynomial trend plus noise. Our operator is also designed to take into account the correlation between the trend and the periodic component. Our extended LS periodogram is however not sufficient to deal with very noisy data sets, and it also exhibits spectral leakage, like the DFT. In the world of regularly sampled, very noisy, time series, *smoothing* techniques can be applied to reduce the variance of the periodogram, after tapering the time series in order to alleviate spectral leakage, see (Harris, 1978). One of them is the WOSA method (Welch, 1967), which consists in segmenting the time series into overlapping segments, tapering them, taking the periodogram on each segment, and finally taking the average of all the periodograms. This technique was transferred to the world of irregularly sampled time series in the work of (Schulz and Stattegger, 1997), where they apply the classical LS periodogram to each tapered segment, and take the average. In this article, we generalize their work by applying the tapered WOSA method to our extended LS periodogram. Moreover, we show that it is preferable to weight the periodogram of each WOSA segment before taking the average, in order to get a reliable representation





of the squared amplitude of the periodic component. This leads us to define the *weighted WOSA periodogram*, that we recommend for most spectral analyses.

The periodogram is often accompanied by a test of significance for the spectral peaks, which relies on the choice of an additive background noise. Two traditional background noises are used in practice. The first one is the Gaussian

white noise, which has a flat power spectral density, and which is a common choice with astronomical data sets, e.g. in (Scargle, 1982) or(Heck et al., 1985). The second one is the Gaussian red noise or Ornstein-Uhlenbeck process, for which the power spectral density is a Lorentzian function centered at frequency zero, and which is a common choice with (paleo)climate time series, e.g. in (Schulz and Mudelsee, 2002) or (Ghil et al., 2002). Arguments in favor of a Gaussian red noise as the background stochastic process for climate time series are given in the influential

paper of (Hasselmann, 1976). Other background noises are also found in geophysics, often under the form of an autoregressive-moving-average (ARMA) process; See (Mudelsee, 2010, p. 60) for an extensive list. In this work, we consider a general class of background noises, which are the continuous autoregressive-moving-average (CARMA) processes, defined in Sect. 3.2. A CARMA(p,q) process is the extension of an ARMA(p,q) process to a continuous time (Tómasson, 2013). Gaussian white noise and Gaussian red noise are particular cases of a Gaussian CARMA

process, i.e. they are a CARMA(0,0) process and a CARMA(1,0) process respectively. Recent advances now allow to accurately estimate the parameters of an irregularly sampled CARMA process, see (Kelly et al., 2014).

Estimating the percentiles of the distribution of the weighted WOSA periodogram of an irregularly sampled CARMA process is the core of this paper. This gives the confidence levels for performing tests of significance at every frequency, i.e. test if the null hypothesis - the time series is a purely stochastic CARMA process - can be rejected (with

some percentage of confidence) or not. We aim at developing a very general approach. Let us enumerate some key points:

1. Estimation of CARMA parameters is performed in a Bayesian framework and relies on state of the art algorithms provided by (Kelly et al., 2014). For the special case of a white noise, we provide an analytical solution.

2. Based on 1, we provide confidence levels computed with Markov chain Monte Carlo (MCMC) methods, that fully take into account the uncertainty on the parameters of the CARMA process, because we work with a *distribution* of values for the CARMA parameters instead of a unique set of values.

3. Alternatively to 2, if we opt for the traditional choice of a unique set of values for the parameters of the CARMA background noise, we develop a theory providing *analytical* confidence levels. Compared to a MCMC-based

approach, the analytical method is more accurate and, if the number of data points is not too high, quicker to compute, especially at high confidence levels, e.g. 99 % or 99.9 %. Computing high levels of confidence is required in some studies, for example in paleoceanography (Kemp, 2016).

4. Confidence levels are provided for any possible choice of the overlapping factor for the WOSA method, extending the traditional 50 % overlapping choice (Schulz and Stattegger, 1997; Schulz and Mudelsee, 2002).


5. Under the case of a white noise background, without WOSA segmentation and without tapering, we define the *F-periodogram* as an alternative to the periodogram. It has the advantage of not requiring any parameter to be estimated.

Finally, we note that spectral power and estimated squared amplitude are no longer the same thing if the time series is irregularly sampled. Both quantities may be of physical interest. We estimate the amplitude of the periodic component with least squares methods, and derive an approximate proportionality between the squared amplitude and the periodogram, from which we deduce the weights for the weighted WOSA periodogram. The estimated signal amplitude also gives access to filtering in a frequency band.

The paper is organized as follows: In Sect. 2, we introduce the notations and recall some basics of algebra. In Sect. 3, we define the model for the data and write the background noise term into a suitable mathematical form. Section 4 starts with some reminders about the Lomb-Scargle periodogram and then extends it to take into account the trend, and a second extension deals with the WOSA tapered case. In Sect. 5, we remind that significance testing is nothing but a statistical hypothesis testing. Under the null hypothesis, we estimate the parameters of the CARMA process and estimate the distribution of the WOSA periodogram, either with Monte-Carlo methods or analytically. In the case of a white noise background, we define the F-periodogram as an alternative to the periodogram. Section 6 aims at computing the amplitude of the periodic component of the signal and the difference between the squared amplitude and the periodogram is explained. Sections 7 and 8 are based on the results of Sect. 6. There, we propose a third extension for the LS periodogram and show how to perform filtering. Section 9 presents an example of analysis on a paleoceanographic time series. Finally, a Python package named WAVEPAL is available to the reader and is presented in Sect. 10.

## 2  Notations and mathematical background

### 2.1  Notations

Let us introduce the notations for the time series. The measurements $X_1, X_2, ..., X_N$ are done at the times $t_1, t_2, ..., t_N$ respectively, and we assume there is no error on the measurements as well as on the times. They are cast into vectors belonging to $\mathbb{R}^N$:

$$|t\rangle = \begin{pmatrix} t_1 \\ t_2 \\ \vdots \\ t_N \end{pmatrix} \quad \text{and} \quad |X\rangle = \begin{pmatrix} X_1 \\ X_2 \\ \vdots \\ X_N \end{pmatrix}. \tag{1}$$

We use here the bra-ket notation, which is common in physics. In $\mathbb{R}^N$, the transpose of $|a\rangle$ is $\langle a|$, i.e. $\langle a|' = |a\rangle$, and in $\mathbb{C}^N$, $\langle a|$ is the conjugate transpose of $|a\rangle$, i.e. $\langle a|^* = |a\rangle$. The inner product of $|a\rangle$ and $|b\rangle$ is $\langle a|b\rangle$.





– Let $A$ be a $(m,n)$ matrix and $B$ be a $(n,m)$ matrix. If $A$ is real, $A'$ denotes its transpose, and if $A$ is complex, $A^*$ denotes its conjugate transpose. The trace of $AB$ is denoted by $\mathrm{tr}(AB)$ and we have $\mathrm{tr}(AB) = \mathrm{tr}(BA)$.

– We use the terminology *Gaussian white noise* or simply *white noise* for a (multivariate) Gaussian random variable with constant mean and covariance matrix $\sigma^2 \mathbb{I}$.

– $|Z\rangle$ always denotes a standard multivariate Gaussian white noise, i.e.

$$|Z\rangle \stackrel{d}{=} \mathcal{N}(0, \mathbb{I}), \tag{2}$$

where $\stackrel{d}{=}$ means "is equal in distribution" and $\mathbb{I}$ is the identity matrix.

– A sequence of independent and identically distributed random variables is denoted by "iid".

## 2.2   Orthogonal projections in $\mathbb{R}^N$

The orthogonal projection on a vector space spanned by the $m$ linearly independent vectors $|a_1\rangle$, ..., $|a_m\rangle$ in $\mathbb{R}^N$ for some $m \in \mathbb{N}_0$ $(m \leq N)$ is

$$P_{\overline{\mathrm{sp}}\{\mathbf{a_1},...,\mathbf{a_m}\}} = V(V'V)^{-1}V', \tag{3}$$

where $\overline{\mathrm{sp}}\{\mathbf{a_1},...,\mathbf{a_m}\}$ is the closed span of those $m$ vectors, i.e. the set of all the linear combinations between them. $V$ is a $(N,m)$ matrix defined by

$$V = \begin{pmatrix} | & & | \\ |a_1\rangle & \dots & |a_m\rangle \\ | & & | \end{pmatrix}. \tag{4}$$

Like for any orthogonal projection, we have the following equalities:

$$P_{\overline{\mathrm{sp}}\{\mathbf{a_1},...,\mathbf{a_m}\}} = P'_{\overline{\mathrm{sp}}\{\mathbf{a_1},...,\mathbf{a_m}\}} = P^2_{\overline{\mathrm{sp}}\{\mathbf{a_1},...,\mathbf{a_m}\}}. \tag{5}$$

The $m$ linearly independent vectors $|a_1\rangle$, ..., $|a_m\rangle$ may be orthonormalized by a Gram-Schmidt procedure, leading to $m$ orthonormal vectors $|b_1\rangle$, ..., $|b_m\rangle$, and the orthogonal projection may then be rewritten as

$$P_{\overline{\mathrm{sp}}\{\mathbf{a_1},...,\mathbf{a_m}\}} = P_{\overline{\mathrm{sp}}\{\mathbf{b_1},...,\mathbf{b_m}\}} = \sum_{k=1}^{m} |b_k\rangle\langle b_k|. \tag{6}$$

Under that form, we see that the above projection has $m$ eigenvalues equal to 1 and $(N-m)$ eigenvalues equal to 0. Let $|c_1\rangle$, ..., $|c_q\rangle$ be $q$ linearly independent vectors in $\mathbb{R}^N$, with $q \leq m$, and such that $\overline{\mathrm{sp}}\{\mathbf{c_1},...,\mathbf{c_q}\} \subseteq \overline{\mathrm{sp}}\{\mathbf{a_1},...,\mathbf{a_m}\}$. Then $(P_{\overline{\mathrm{sp}}\{\mathbf{a_1},...,\mathbf{a_m}\}} - P_{\overline{\mathrm{sp}}\{\mathbf{c_1},...,\mathbf{c_q}\}})$ is an orthogonal projection on $\overline{\mathrm{sp}}\{\mathbf{c_1},...,\mathbf{c_q}\} \cap \overline{\mathrm{sp}}\{\mathbf{a_1},...,\mathbf{a_m}\}^{\perp}$, and

$$P_{\overline{\mathrm{sp}}\{\mathbf{a_1},...,\mathbf{a_m}\}} P_{\overline{\mathrm{sp}}\{\mathbf{c_1},...,\mathbf{c_q}\}} = P_{\overline{\mathrm{sp}}\{\mathbf{c_1},...,\mathbf{c_q}\}} P_{\overline{\mathrm{sp}}\{\mathbf{a_1},...,\mathbf{a_m}\}} = P_{\overline{\mathrm{sp}}\{\mathbf{c_1},...,\mathbf{c_q}\}}. \tag{7}$$

Moreover, for any vector $|Y\rangle \in \mathbb{R}^N$, we have

$$||(P_{\overline{\mathrm{sp}}\{\mathbf{a_1},...,\mathbf{a_m}\}} - P_{\overline{\mathrm{sp}}\{\mathbf{c_1},...,\mathbf{c_q}\}})|Y\rangle||^2 = ||P_{\overline{\mathrm{sp}}\{\mathbf{a_1},...,\mathbf{a_m}\}}|Y\rangle||^2 - ||P_{\overline{\mathrm{sp}}\{\mathbf{c_1},...,\mathbf{c_q}\}}|Y\rangle||^2. \tag{8}$$

We recommend the book (Brockwell and Davis, 1991) for more details.



## 2.3 Quantifying the irregularity of the sampling

The biggest time step for which $t_1$, ..., $t_N$, are a subsample of a regularly sampled time series is the greatest common divisor[1] (GCD) of all the time steps of $|t\rangle$. In formulas:

$$\Delta t_{\mathrm{GCD}} = \mathrm{GCD}(\Delta t_1, ..., \Delta t_{N-1}), \text{ where } \Delta t_k = t_{k+1} - t_k \ \forall k \in \{1, ..., N-1\}, \tag{9}$$

and

$$\forall k \in \{1, ..., N\}, \ \exists m \in \mathbb{Z} \text{ s.t. } t_k = m \Delta t_{\mathrm{GCD}}, \tag{10}$$

where $\mathbb{Z}$ denotes the space of integer numbers. Quantifying the irregularity of the sampling is then straightforward. We define

$$r_t = 100 \frac{(N-1)\Delta t_{\mathrm{GCD}}}{t_N - t_1}. \tag{11}$$

This ratio is between $0\,\%$ and $100\,\%$, the latter value being reached with regularly sampled time series.

## 3 The model for the data

### 3.1 Definition

A suitable and general enough model to analyze the periodicity at frequency $f = \frac{2\pi}{\omega}$ is:

$$|X\rangle = |\mathrm{Trend}\rangle + E_\omega \cos(\omega|t\rangle + \phi_\omega) + |\mathrm{Noise}\rangle$$
$$= |\mathrm{Trend}\rangle + A_\omega |c_\omega\rangle + B_\omega |s_\omega\rangle + |\mathrm{Noise}\rangle, \tag{12}$$

with $A_\omega = E_\omega \cos(\phi_\omega)$, $B_\omega = -E_\omega \sin(\phi_\omega)$, $E_\omega^2 = A_\omega^2 + B_\omega^2$, $|c_\omega\rangle = \cos(\omega|t\rangle) = [\cos(\omega t_1), ..., \cos(\omega t_N)]'$ and $|s_\omega\rangle = \sin(\omega|t\rangle) = [\sin(\omega t_1), ..., \sin(\omega t_N)]'$.

### 3.2 The background noise

#### 3.2.1 Definition of a CARMA process

The background noise term, $|\mathrm{Noise}\rangle$, considered in this paper is a zero-mean stationary Gaussian continuous autoregressive-moving-average (CARMA) process sampled at the times of $|t\rangle$. As explained in the following, it generalizes traditional background noises used in geophysics.

A CARMA(p,q) process is simply the extension of an ARMA(p,q) process to a continuous time[2]. A zero-mean

---

[1]The GCD is usually defined on the integers, but we can extend it to rational numbers. In practice, $t_1$, ..., $t_N$ come from measurements with a finite precision and are thus rational numbers.

[2]A CARMA(p,q) process sampled at the times of an infinite regularly sampled time series is an ARMA(p,q) process.




CARMA(p,q) process $y(t)$ is the solution of the following stochastic differential equation:

$$\frac{\mathrm{d}^p y(t)}{\mathrm{d}t^p} + \alpha_{p-1}\frac{\mathrm{d}^{p-1} y(t)}{\mathrm{d}t^{p-1}} + ... + \alpha_0 y(t) = \beta_q \frac{\mathrm{d}^q \varepsilon(t)}{\mathrm{d}t^q} + \beta_{q-1}\frac{\mathrm{d}^{q-1}\varepsilon(t)}{\mathrm{d}t^{q-1}} + ... + \varepsilon(t), \tag{13}$$

where $\varepsilon(t)$ is a continuous-time white noise process with zero mean and variance $\sigma^2$. The parameters $\alpha_0, ... , \alpha_{p-1}$ are the autoregressive coefficients, and the parameters $\beta_1, ..., \beta_q$ are the moving average coefficients. $\alpha_p = \beta_0 = 1$ by definition. When $p > 0$, the process is stationary only if $q < p$ and the roots $r_1, ..., r_p$ of

$$\sum_{k=0}^{p} \alpha_k z^k = 0, \tag{14}$$

have negative real parts. We follow here the definitions and conventions of (Kelly et al., 2014). More details and additional references can be found therein. In practice, only CARMA processes of low order are useful in our framework, typically, $(p,q)$=(0,0), (1,0), (2,0), (2,1), since at higher order, they often exhibit dominant spectral peaks, see (Kelly et al., 2014), which is not what we want as a model for the spectral background. Indeed, on the basis of our model, Eq. (12), it is desirable that the spectral peaks come from the deterministic cosine and sine components. We now consider two useful particular cases of a CARMA process before analyzing the general case.

### 3.2.2 Gaussian white noise

When $p = 0$ and $q = 0$, the process reduces to a white noise, normally distributed with zero-mean and variance $\sigma^2$. The $|\mathrm{Noise}\rangle$ term in Eq. (12) is then simply

$$|\mathrm{Noise}\rangle = \sigma|Z\rangle = K|Z\rangle, \tag{15}$$

with $K = \sigma\mathbb{I}$.

### 3.2.3 Gaussian red noise

When $p = 1$ and $q = 0$, the CARMA(1,0) or CAR(1) process is an Ornstein-Uhlenbeck process or red noise (Uhlenbeck and Ornstein, 1930), which is quite of interest in geophysical and other applications (Mudelsee, 2010). Since we work with a discrete time series, it is necessary to find the solution of Eq. (13) at $t_1, ..., t_N$. This is done by integrating that equation between consecutive times, i.e. from $t_{i-1}$ to $t_i$ $\forall i \in \{2,...,N\}$. The components of the $|\mathrm{Noise}\rangle$ vector are then:

$$y(t_1) \overset{d}{=} \mathcal{N}(0, \frac{\sigma^2}{2\alpha}),$$

$$y(t_i) = \rho_i y(t_{i-1}) + \eta_i \qquad \forall i \in \{2,...,N\}, \tag{16}$$

where

$$\rho_i = \exp(-\alpha(t_i - t_{i-1})) \quad \text{and} \quad \eta_i \overset{d}{=} \mathcal{N}(0, \frac{\sigma^2}{2\alpha}(1 - \rho_i^2)). \tag{17}$$





See (Robinson, 1977) and (Gardiner, 2009) for more details. The requirement on stationarity, Eq. (14), imposes $\alpha > 0$. The generated time series has a constant mean equal to zero and a constant variance equal to $\frac{\sigma^2}{2\alpha}$, and the process is thus stationary. The $|\text{Noise}\rangle$ term in Eq. (12) can also be written under a matrix form:

$$|\text{Noise}\rangle = K|Z\rangle, \tag{18}$$

where $K$ is a (N,N) lower triangular matrix whose elements are

$$K_{i,j} = \sqrt{\frac{\sigma^2}{2\alpha}}\sqrt{1-\rho_j^2}\exp(-\alpha(t_i-t_j)) \quad \forall j \leq i, \tag{19}$$

where we define $\rho_1 = 0$. This matrix form is used in Sect. 5.3.3.

Note that, if the time series is regularly sampled, $\rho$ is a constant and Eq. (16) becomes the equation of a finite-length AR(1) process, which is stationary since $\alpha > 0$ implies $\rho < 1$.

### 3.2.4 The general Gaussian CARMA noise

The solution of Eq. (13) at the time $t_n$ $(n = 2, ..., N)$, that we denote by $y_n$, is

$$y_n = \langle b | w_n \rangle,$$

where $|w_n\rangle = \exp(A(t_n - t_{n-1}))|w_{n-1}\rangle + |\eta_n\rangle, \tag{20}$

and where $|b\rangle = [\beta_0, \beta_1, ..., \beta_q, 0, ..., 0]'$ is a vector of length $p$, and

$$A = \begin{pmatrix} 0 & 1 & 0 & \dots & 0 \\ 0 & 0 & 1 & \dots & 0 \\ \vdots & \vdots & \vdots & \ddots & \vdots \\ 0 & 0 & 0 & \dots & 1 \\ -\alpha_0 & -\alpha_1 & -\alpha_2 & \dots & -\alpha_{p-1} \end{pmatrix}. \tag{21}$$

$|\eta_n\rangle$ follows a multivariate normal distribution with zero mean and covariance matrix $C_n$ given by

$$C_n = \int\limits_0^{t_n-t_{n-1}} \mathrm{d}t \exp(At)|e\rangle\langle e|\exp(A't), \tag{22}$$

where $|e\rangle = [0, 0, ..., 0, \sigma]'$. The above formula requires the computation of matrix exponentials and numerical integration. This can be avoided by diagonalizing matrix A, with $A = UDU^{-1}$. D is a diagonal matrix with diagonal elements given by the roots of Eq. (14):

$$D_{kk} = r_k \quad \forall k \in 1, ..., p, \tag{23}$$

and $U$ is a Vandermonde matrix, with

$$U_{lk} = r_k^{l-1} \quad \forall l, k \in 1, ..., p. \tag{24}$$



Now, by defining $|\tilde{w}_n\rangle = U^{-1}|w_n\rangle$, we get

$$y_n = \langle b|U|\tilde{w}_n\rangle, \tag{25a}$$

where $|\tilde{w}_n\rangle = \Lambda_n|\tilde{w}_{n-1}\rangle + |\tilde{\eta}_n\rangle.$ (25b)

The matrix exponential $\exp(A(t_n - t_{n-1}))$ has been transformed into $\Lambda_n = U^{-1}\exp(A(t_n - t_{n-1}))U$ which is simply a diagonal matrix with elements $\Lambda_{n,kk} = \exp(r_k(t_n - t_{n-1}))$. The covariance matrix of $|\tilde{\eta}_n\rangle$, that we write $\Sigma_n$, also takes a relatively simple form:

$$\Sigma_{n,kl} = -\sigma^2 \frac{\kappa_k \kappa_l^*}{(r_k + r_l^*)}(1 - \exp((r_k + r_l^*)(t_n - t_{n-1}))) \quad \forall k,l \in \{1,...,p\}, \tag{26}$$

which is a Hermitian matrix, and where $|\kappa\rangle$ is the last column of $U^{-1}$. The initial condition $y_1$ is determined by imposing stationarity, which is fulfilled only if $|w_1\rangle$ has a zero mean and a covariance matrix $V$ whose elements are

$$V_{kl} = -\sigma^2 \sum_{m=1}^{p} \frac{r_m^{k-1}(-r_m)^{l-1}}{2\mathrm{Re}\{r_m\}\prod_{s=1,s\neq m}^{p}(r_s - r_m)(r_s^* + r_m)} \quad \forall k,l \in \{1,...,p\}. \tag{27}$$

Stationarity implies that the process $y(t)$ has a zero mean and variance $\langle b|V|b\rangle \; \forall t$. All the above formulas and how to get them can be found in (Kelly et al., 2014), (Jones and Ackerson, 1990) and (Gardiner, 2009).

Generation of a CARMA$(p,q)$ process can be performed with the Kalman filter since Eq. (25b) and (25a) are nothing but the state and measurement equations respectively. See (Kelly et al., 2014) for more details. Alternatively, $|y\rangle$

can be written under a matrix form as in Eq. (18). Matrix formalism is useful in Sect. 5.3.3. Let us start with Eq. (25b):

$$|\tilde{w}_n\rangle = \Lambda_n|\tilde{w}_{n-1}\rangle + U^{-1}|\eta_n\rangle. \tag{28}$$

The covariance matrix of $|\eta_n\rangle$, $C_n = U\Sigma U^*$, is of course real symmetric and positive semi-definite. We thus have the following Schur decomposition:

$$C_n = Q_n Q'_n, \tag{29}$$

where $Q_n$ is a real matrix. Consequently,

$$
\begin{aligned}
|\tilde{w}_n\rangle &= \Lambda_n|\tilde{w}_{n-1}\rangle + U^{-1}Q_n|\varepsilon_n\rangle \\
&= \Lambda_n\Lambda_{n-1}|\tilde{w}_{n-2}\rangle + \Lambda_n U^{-1}Q_{n-1}|\varepsilon_{n-1}\rangle + U^{-1}Q_n|\varepsilon_n\rangle \\
&= ... \\
&= \sum_{i=2}^{n}(\prod_{l=i+1}^{n}\Lambda_l)U^{-1}Q_i|\varepsilon_i\rangle + \prod_{l=2}^{n}\Lambda_l|\tilde{w}_1\rangle,
\end{aligned}
\tag{30}
$$

where $|\varepsilon_1\rangle, ..., |\varepsilon_n\rangle$ are iid standard Gaussian white noises. The product of the $\Lambda$'s can be simplified. Its diagonal

elements are:

$$(Y_{in})_{jj} := (\prod_{l=i+1}^{n}\Lambda_l)_{jj} = \exp(r_j(t_n - t_i)). \tag{31}$$





As stated above, $|w_1\rangle$ follows a multivariate normal distribution with zero mean and covariance matrix $V$. We can use again the Schur decomposition to write $V = WW'$, where $W$ is a real matrix, yielding

$$|\tilde{w}_n\rangle = \sum_{i=2}^{n} Y_{in}U^{-1}Q_i|\varepsilon_i\rangle + Y_{1n}U^{-1}W|\varepsilon_1\rangle$$

$$= \sum_{i=1}^{n} P_{in}|\varepsilon_i\rangle, \tag{32}$$

with $P_{1n} = Y_{1n}U^{-1}W$ and $P_{in} = Y_{in}U^{-1}Q_i$ for $i > 1$. The CARMA process at time $t_n$ is then given by

$$y_n = \langle b|U|\tilde{w}_n\rangle$$

$$= \sum_{i=1}^{n} \langle b|U|P_{in}|\varepsilon_i\rangle. \tag{33}$$

Finally, the $|\text{Noise}\rangle$ term in Eq. (12) is

$$|\text{Noise}\rangle = |y\rangle = \begin{pmatrix} \langle b|U|P_{11} & \langle 0| & \dots & \dots & \langle 0| \\ \langle b|U|P_{12} & \langle b|U|P_{22} & \langle 0| & \dots & \langle 0| \\ & & \ddots & & \\ & & & \ddots & \\ \langle b|U|P_{1N} & \langle b|U|P_{2N} & \dots & \dots & \langle b|U|P_{NN} \end{pmatrix} \begin{pmatrix} |\varepsilon_1\rangle \\ |\varepsilon_2\rangle \\ \vdots \\ |\varepsilon_N\rangle \end{pmatrix} = K|Z\rangle, \tag{34}$$

where $K$ is a $(N, N \times p)$ real matrix and $|Z\rangle$ has a length $N \times p$. Matrix $K$ is triangular if $p = 1$, which is the particular case treated in Sect. 3.2.3.

### 3.3 The trend

The model for the trend must be as general as possible and must be convenient for a formalism based on orthogonal projections (see Sect. 4). This is the reason why we choose a polynomial trend of some degree $m$:

$$|\text{Trend}\rangle = \sum_{k=0}^{m} \gamma_k |t^k\rangle, \text{ where } |t^k\rangle = [t_1^k, ..., t_N^k]'. \tag{35}$$

## 4 Periodogram & relatives

### 4.1 Lomb-Scargle periodogram

Consider the orthogonal projection of the data $|X\rangle$ onto the vector space spanned by the vectors cosine and sine, defined by $|c_\omega\rangle = \cos(\omega|t\rangle)$ and $|s_\omega\rangle = \sin(\omega|t\rangle)$. The periodogram at the frequency $f = \frac{\omega}{2\pi}$ is defined as the squared norm of that projection:

$$||P_{\overline{\text{sp}}\{\mathbf{c}_\omega, \mathbf{s}_\omega\}}|X\rangle||^2. \tag{36}$$


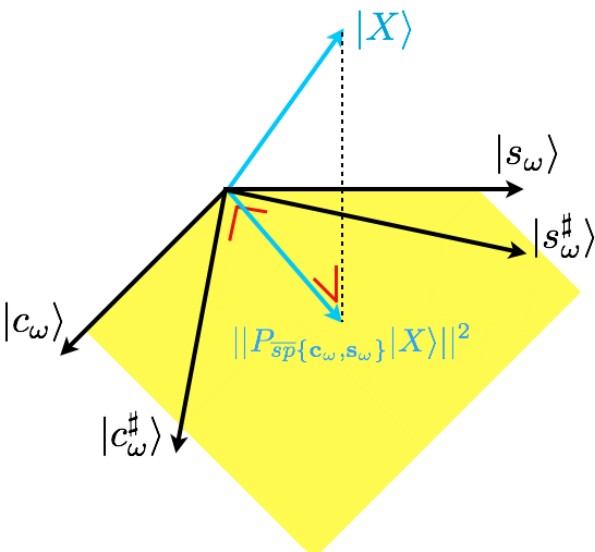

**Figure 1.** Schematic view of the linear rescaling in $\mathbb{R}^N$ leading to the Lomb-Scargle formulas. In yellow is drawn a subset of $\overline{\mathrm{sp}}\{\mathbf{c}_\omega, \mathbf{s}_\omega\}$. A span is invariant under linear combinations of its vectors. The dashed line corresponds to the minimal euclidean distance between the data and $\overline{\mathrm{sp}}\{\mathbf{c}_\omega, \mathbf{s}_\omega\}$.

When the time series is regularly sampled with a constant time step $\Delta t$, and if we only consider the Fourier angular frequencies, $\omega_k = \frac{2\pi k}{N \Delta t}$ ($k = 0, ..., N-1$), the periodogram defined above is equal to the squared norm of the discrete Fourier transform (DFT) of real signals.

Now, rescale $|c_\omega\rangle$ and $|s_\omega\rangle$ such that they are orthonormal. This can be done by defining

$$5 \quad |c_\omega^\sharp\rangle = \frac{\cos(\omega|t\rangle - \beta_\omega)}{\sqrt{\Sigma_{i=1}^N \cos^2(\omega t_i - \beta_\omega)}}, \qquad |s_\omega^\sharp\rangle = \frac{\sin(\omega|t\rangle - \beta_\omega)}{\sqrt{\Sigma_{i=1}^N \sin^2(\omega t_i - \beta_\omega)}}, \tag{37}$$

where $\beta_\omega$ is the solution of

$$\tan(2\beta_\omega) = \frac{\Sigma_{i=1}^N \sin(2\omega t_i)}{\Sigma_{i=1}^N \cos(2\omega t_i)}. \tag{38}$$

The spanned vector space naturally remains unchanged (see Fig. 1). These formulas are nothing but the Lomb-Scargle formulas (Scargle, 1982, Eq. (10)). The periodogram is now

$$10 \quad ||P_{\overline{\mathrm{sp}}\{\mathbf{c}_\omega, \mathbf{s}_\omega\}}|X\rangle||^2 = \langle c_\omega^\sharp | X\rangle^2 + \langle s_\omega^\sharp | X\rangle^2. \tag{39}$$

Note that, for any signal $|X\rangle$,

$$0 \leq \frac{||P_{\overline{\mathrm{sp}}\{\mathbf{c}_\omega, \mathbf{s}_\omega\}}|X\rangle||^2}{\langle X | X\rangle} \leq 1, \tag{40}$$





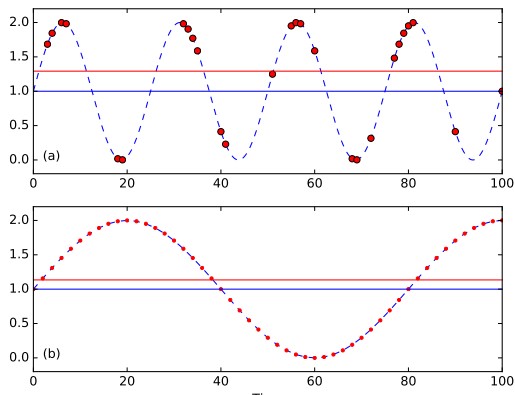

**Figure 2.** Signal average and sampling. (a) The continuous signal is in dashed blue and it is irregularly sampled at red dots. The continuous signal oscillates around 1 (blue line), which does not correspond to the average of the sampled signal (red line). (b) Same as (a) with a regularly sampled signal.

and this is equal to 1 if $|X\rangle = A|c_\omega\rangle + B|s_\omega\rangle$.

Some properties of the LS periodogram are presented in appendix A. Here and for the rest of the article, the frequency $f = \omega/2\pi$ is considered as a continuous variable.

### 4.2 Periodogram and mean

The LS periodogram applies well to data which can be modeled as

$$|X\rangle = A_\omega|c_\omega\rangle + B_\omega|s_\omega\rangle + |\text{Noise}\rangle. \tag{41}$$

However, the periodic components may not necessarily oscillate around zero, and a better model is

$$|X\rangle = \mu|t^0\rangle + A_\omega|c_\omega\rangle + B_\omega|s_\omega\rangle + |\text{Noise}\rangle, \tag{42}$$

where $|t^0\rangle = [1, 1, ..., 1]'$. Substracting the average of the data is then often done before applying the LS periodogram.

But that mere operation implicitly assumes that $\langle t^0|c\rangle = \langle t^0|s\rangle = 0$, which is not necessarily the case. In other words, the data average is not necessarily equal to $\mu$, the process mean. Fig. 2a illustrates that fact. Note that this discrepancy occurs in regularly sampled data as well, at non-Fourier frequencies, i.e. when $N\Delta t$ is not a multiple of the probing period. See Fig. 2b. In order to deal with the mean in a suitable way, we define the periodogram as

$$||(P_{\overline{\text{sp}}\{t^0, c_\omega, s_\omega\}} - P_{\overline{\text{sp}}\{t^0\}})|X\rangle||^2. \tag{43}$$

Formula (43) is taken from (Brockwell and Davis, 1991), (Ferraz-Mello, 1981) or (Heck et al., 1985); equivalence between them is shown in appendix B. $[P_{\overline{\text{sp}}\{t^0, c_\omega, s_\omega\}} - P_{\overline{\text{sp}}\{t^0\}}]$ is also an orthogonal projection. A simple example will





justify the principle. Consider the following purely deterministic mono-periodic signal with $N$ data points:

$$|Y\rangle = \mu|t^0\rangle + A|c_\omega\rangle + B|s_\omega\rangle = V_3|\Phi\rangle, \text{ with } \begin{pmatrix} | & | & | \\ |t^0\rangle & |c_\omega\rangle & |s_\omega\rangle \\ | & | & | \end{pmatrix}, \text{ and } |\Phi\rangle = \begin{pmatrix} \mu \\ A \\ B \end{pmatrix}. \tag{44}$$

The projection at $\omega$ is

$$\begin{aligned}
(P_{\overline{\mathrm{sp}}\{\mathbf{t^0},\mathbf{c}_\omega,\mathbf{s}_\omega\}} - P_{\overline{\mathrm{sp}}\{\mathbf{t^0}\}})|Y\rangle &= (\mathbb{I} - P_{\overline{\mathrm{sp}}\{\mathbf{t^0}\}})P_{\overline{\mathrm{sp}}\{\mathbf{t^0},\mathbf{c}_\omega,\mathbf{s}_\omega\}}|Y\rangle \\
&= (\mathbb{I} - P_{\overline{\mathrm{sp}}\{\mathbf{t^0}\}})V_3|\Phi\rangle \\
&= |Y\rangle - P_{\overline{\mathrm{sp}}\{\mathbf{t^0}\}}|Y\rangle \\
&= A|c_\omega\rangle + B|s_\omega\rangle - \frac{\langle t^0|c_\omega\rangle}{\langle t^0|t^0\rangle}A|t^0\rangle - \frac{\langle t^0|s_\omega\rangle}{\langle t^0|t^0\rangle}B|t^0\rangle.
\end{aligned} \tag{45}$$

We see that it is invariant with respect to $\mu$, and we find back the signal minus its average. We thus have

$$||(P_{\overline{\mathrm{sp}}\{\mathbf{t^0},\mathbf{c}_\omega,\mathbf{s}_\omega\}} - P_{\overline{\mathrm{sp}}\{\mathbf{t^0}\}})|Y\rangle||^2 = N\,\mathrm{Var}(|Y\rangle), \tag{46}$$

which is a result similar to what we get with regularly sampled data and the DFT[3].

Now, we do a Gram-Schmidt orthonormalization like in (Ferraz-Mello, 1981), in order to simplify formula (43). To this end, we define the three orthonormal vectors $|h_0\rangle = |t^0\rangle/|||t^0\rangle||$, $|h_1\rangle$ and $|h_2\rangle$ satisfying

$$\overline{\mathrm{sp}}\{\mathbf{t^0},\mathbf{c}_\omega,\mathbf{s}_\omega\} = \overline{\mathrm{sp}}\{\mathbf{h_0},\mathbf{h_1},\mathbf{h_2}\}. \tag{47}$$

Consequently,

$$P_{\overline{\mathrm{sp}}\{\mathbf{t^0},\mathbf{c}_\omega,\mathbf{s}_\omega\}} - P_{\overline{\mathrm{sp}}\{\mathbf{t^0}\}} = |h_1\rangle\langle h_1| + |h_2\rangle\langle h_2|, \tag{48}$$

and

$$||(P_{\overline{\mathrm{sp}}\{\mathbf{t^0},\mathbf{c}_\omega,\mathbf{s}_\omega\}} - P_{\overline{\mathrm{sp}}\{\mathbf{t^0}\}})|X\rangle||^2 = \langle h_1|X\rangle^2 + \langle h_2|X\rangle^2. \tag{49}$$

Note that, for any signal $|X\rangle$ with $N$ data points, we have

$$0 \le \frac{||(P_{\overline{\mathrm{sp}}\{\mathbf{t^0},\mathbf{c}_\omega,\mathbf{s}_\omega\}} - P_{\overline{\mathrm{sp}}\{\mathbf{t^0}\}})|X\rangle||^2}{N\,\mathrm{Var}(|X\rangle)} \le 1, \tag{50}$$

and this is equal to 1 for a signal given by $|X\rangle = \mu|t^0\rangle + A|c_\omega\rangle + B|s_\omega\rangle$.

---

[3]If we have $|Y\rangle = \mu|t^0\rangle + A|e_\omega\rangle$, where $|e_\omega\rangle = \exp(i2\pi\omega|t\rangle)$ and $\omega$ being a Fourier frequency, then $||DFT(|Y\rangle)||^2 = ||P_{\overline{\mathrm{sp}}\{\mathbf{e}_\omega\}}|Y\rangle||^2 = N\,||A||^2 = N\,\mathrm{Var}(|Y\rangle)$. Var is here the biased variance, which is defined as the squared norm of the signal minus its average value, and divided by $N$.





### 4.3 Periodogram and a polynomial trend

If we want to work with the full model, Eq. (12), which has a polynomial trend of degree $m$, we can naturally extend the result of Sect. 4.2 and work with

$$||(P_{\overline{sp}\{t^0, t^1, ..., t^m, c_\omega, s_\omega\}} - P_{\overline{sp}\{t^0, t^1, ..., t^m\}})|X\rangle||^2 = \langle h_{m+1}|X\rangle^2 + \langle h_{m+2}|X\rangle^2, \tag{51}$$

where $|h_{m+1}\rangle$ and $|h_{m+2}\rangle$ are determined from a Gram-Schmidt orthonormalization starting with the orthonormalization of $|t^0\rangle$, ..., $|t^m\rangle$.

It may happen that, for large $m$, the correlation matrix in the formula of orthogonal projection be singular. In that case, two options, less optimal, are possible: reduce the degree $m$, or perform the detrending before the spectral analysis, for example with a moving average.

Similarly to Sect. 4.2, we have, for any signal $|X\rangle$,

$$0 \le \frac{||(P_{\overline{sp}\{t^0, t^1, ..., t^m, c_\omega, s_\omega\}} - P_{\overline{sp}\{t^0, t^1, ..., t^m\}})|X\rangle||^2}{|||X\rangle - P_{\overline{sp}\{t^0, ..., t^m\}}|X\rangle||^2} \le 1, \tag{52}$$

and this is equal to 1 for a signal given by $|X\rangle = \sum_{k=0}^{m} \gamma_k |t^k\rangle + A|c_\omega\rangle + B|s_\omega\rangle$. Finally, we have a result similar to Eq. (45), in the sense that the projection given in Eq. (51) is invariant with respect to the parameters of the trend (but it naturally depends on the choice of the *degree* m).

### 4.4 Tapering the periodogram

A finite length signal can be seen as an infinite length signal multiplied by a rectangular window. This implies, among others, that a mono-periodic signal will have a periodogram characterized by a peak of finite width, with possibly large sidelobes, instead of a Dirac delta function. This is called *spectral leakage*.

The phenomenon has been deeply studied in the case of regularly sampled data. Leakage may be controlled by
choosing alternatives to the default rectangular window. This is called *windowing* or *tapering*. See (Harris, 1978) for an extensive list of windows. They all share the property to vanish at the borders of the time series.

In the case of irregularly sampled data, building windows for controlling the leakage is a much more challenging task. Even with the default rectangular window, leakage is very irregular, data and frequency dependent, due to the long-range correlations in frequency between the vectors on which we do the projection. To our knowledge,
no general and stable solution for that issue is available in the literature. We thus recommend to use the default rectangular window, i.e. do no tapering, if $r_t$, defined in Eq. (11), is small, and to use simple windows, like the $sin^2$ or the Gaussian window, for moderately irregularly sampled data ($r_t$ greater than 80 % or 90 %). With tapering, formula (51) becomes

$$||(P_{\overline{sp}\{t^0, t^1, ..., t^m, Gc_\omega, Gs_\omega\}} - P_{\overline{sp}\{t^0, t^1, ..., t^m\}})|X\rangle||^2, \tag{53}$$





where $G$ is a frequency-independent diagonal matrix, which is used to weight the sine and cosine vectors. For example, with a $\sin^2$ window, also called Hanning window, we have

$$G_{kk} = \sin^2\left(\frac{\pi\,(t_k - t_1)}{t_N - t_1}\right) \qquad \forall k \in \{1, ..., N\}. \tag{54}$$

## 4.5 Smoothing the periodogram with the WOSA method

### 4.5.1 The consistency problem

Besides spectral leakage, another issue with the periodogram is consistency. Indeed, for regularly sampled time series, the periodogram is known not to be a consistent estimator of the true spectrum as the number of data points tends to infinity, see (Brockwell and Davis, 1991, Chap. 10). Another view of the problem is that the periodogram remains very noisy whatever the number of data points we have at our disposal. Smoothing procedures are therefore required, and some methods are available in the literature (see (Walden, 2000) for a unified view). Among them, two techniques are traditionally used: multitaper methods (MTM), developed by (Thomson, 1982) and (Riedel and Sidorenko, 1995), and Welch overlapping segment averaging (WOSA) method (Welch, 1967).

Multitaper methods are certainly not generalizable to the case of irregularly sampled data, except in very specific cases that are not of interest in geophysics, like in (Bronez, 1988), which deals with band-limited signals, useful in the field of the telecommunications, or (Fodor and Stark, 2000), which considers regularly sampled time series with some gaps, useful for time series with a ratio $r_t$, defined in Eq. (11), close to 100. We will then use the WOSA method applied to the LS periodogram, like in (Schulz and Stattegger, 1997; Schulz and Mudelsee, 2002), or to its relatives (formulas (43), (51), or the most general (53)).

### 4.5.2 Principle of the WOSA method

### 4.5.3 Trendless time series

The time series is divided into overlapping segments. The tapered LS periodogram is computed on every segment, and the WOSA periodogram is the average of all these tapered periodograms. This technique relies on the fact that the signal is stationary, as always in spectral analysis[4]. The length of the segments and the overlapping factor need to be chosen depending on how much we want to reduce the variance of the noise. As a general rule, shortening the segments will decrease the frequency resolution. Consequently, there is always a trade-off between the frequency resolution and the variance reduction.

For regularly sampled data, each segment of fixed length has the same number of data points. In the irregularly sampled case, it is not anymore the case and we have two options.

---

[4]Basically, the *spectrum* cannot be defined without that hypothesis. See the Wiener-Khinchin theorem in e.g. (Priestley, 1981, Chap. 4)



1. Take segments with a fixed number of points and thus a variable length. In the non-tapered case, the periodogram on each segment provides deterministic peaks (coming from the deterministic sin/cos components) with more or less the same height. But variable length segments will give deterministic peaks of variable width.

2. Take segments of fixed length but with a variable number of data points. The periodogram on each segment provides deterministic peaks with more or less the same width, except if there is a big gap at the beginning or at the end of the segment, such that its effective length is reduced. But they will have variable height since the number of data points is not constant.

We judge it is better to have peaks with similar width on each segment when averaging the periodograms in a frequency band. Consequently, we recommend the second option. An example of WOSA segmentation is shown on Fig. 8a.

### 4.5.4 Time series with a trend

The only difference with the previous case is that, for each segment, we consider the projection on $t^0, ..., t^m$ jointly with the tapered cosine and sine components. Formula (53) is applied to each segment with $|Gc_\omega\rangle$ and $|Gs_\omega\rangle$ localized on the WOSA segment, but $|t^0\rangle, ..., |t^m\rangle$ are taken on the full length of the time series, because the trend is the one of the whole time series.

### 4.5.5 The WOSA periodogram in formulas

Two parameters are required: the length of WOSA segments, $D$, and the overlapping factor, $\beta \in [0,1[. \; \beta = 0$ when there is no overlap. We denote by $Q$ the number of WOSA segments, which is equal to

$$Q = \left\lfloor \frac{t_N - t_1 - D}{(1 - \beta)D} \right\rfloor + 1, \tag{55}$$

where $\lfloor \rfloor$ is the floor function. Because of the rounding, $D$ must be adjusted afterwards:

$$D = \frac{t_N - t_1}{1 + (1 - \beta)(Q - 1)}. \tag{56}$$

Define $\tau_q$ to be the starting time of the $q^{\text{th}}$ segment ($q \in \{1, ..., Q\}$). Note that $\tau_q$ is not necessarily equal to one of the components of $|t\rangle$. It follows that

$$\tau_q = t_1 + (1 - \beta)(q - 1)D \quad q = 1, ..., Q. \tag{57}$$

The WOSA periodogram is then

$$||P_{\text{WOSA}}(\omega)|X\rangle||^2 = \frac{1}{Q} \sum_{q=1}^{Q} ||(P_{\overline{\text{sp}}\{\mathbf{t^0, t^1, ..., t^m, G_q c_{\omega,q}, G_q s_{\omega,q}}\}} - P_{\overline{\text{sp}}\{\mathbf{t^0, t^1, ..., t^m}\}})|X\rangle||^2$$

$$= \frac{1}{Q} \sum_{q=1}^{Q} \langle X|(P_{\overline{\text{sp}}\{\mathbf{t^0, t^1, ..., t^m, G_q c_{\omega,q}, G_q s_{\omega,q}}\}} - P_{\overline{\text{sp}}\{\mathbf{t^0, t^1, ..., t^m}\}})|X\rangle. \tag{58}$$



Note that the sum of these orthogonal projections is not anymore an orthogonal projection. $|G_q c_{\omega,q}\rangle$ and $|G_q s_{\omega,q}\rangle$ are the tapered cosine and sine on the $q^{\text{th}}$ segment. For example, with the Hanning ($\sin^2$) window,

$$(G_q c_{\omega,q})_k = g_q(t_k)\cos(\omega(t_k - \tau_q)), \qquad (G_q s_{\omega,q})_k = g_q(t_k)\sin(\omega(t_k - \tau_q)), \tag{59}$$

where

$$g_q(t_k) = \sin^2\left(\frac{\pi\,(t_k - \tau_q)}{D}\right) \quad \text{if} \quad 0 \leq (t_k - \tau_q) \leq D$$

$$= 0 \quad \text{otherwise.} \tag{60}$$

It may be shown that $\overline{\text{sp}}\{\mathbf{t^0}, \mathbf{t^2}, ..., \mathbf{t^m}, \mathbf{G_q c_{\omega,q}}, \mathbf{G_q s_{\omega,q}}\}$ is invariant with the variable $\tau_q$ appearing in the cosine and sine terms, so that we can impose $\tau_q = 0 \,\forall q$ inside the cosine and sine terms.

In formula (58), for each orthogonal projection, we apply a Gram-Schmidt orthonormalization (similarly to Sect. 4.3):

$$||P_{\text{WOSA}}(\omega)|X\rangle||^2 = \frac{1}{Q}\sum_{q=1}^{Q}\left(\langle X|h_{1,q}(\omega)\rangle\langle h_{1,q}(\omega)|X\rangle + \langle X|h_{2,q}(\omega)\rangle\langle h_{2,q}(\omega)|X\rangle\right), \tag{61}$$

where, for each $q$, $|h_{1,q}(\omega)\rangle$ and $h_{2,q}(\omega)\rangle$ are orthonormal. We are now able to write the WOSA periodogram under a simple matrix form:

$$||P_{\text{WOSA}}(\omega)|X\rangle||^2 = \langle X|M_{2,\omega}M'_{2,\omega}|X\rangle, \tag{62}$$

where

$$M_{2,\omega} = \frac{1}{\sqrt{Q}}\begin{pmatrix} | & | & & | & | \\ |h_{1,1}(\omega)\rangle & |h_{2,1}(\omega)\rangle & \cdots & |h_{1,Q}(\omega)\rangle & |h_{2,Q}(\omega)\rangle \\ | & | & & | & | \end{pmatrix}. \tag{63}$$

### 4.5.6 Practical considerations

First, note that the Gram-Schmidt orthonormalization process requires at least $m+3$ data points. WOSA segments with less than $m+3$ points must therefore be ignored in the average of the periodograms.

Second, as we want to get deterministic peaks with more or less the same width on every segment, a WOSA segment is kept in the average if the data cover some percentage of its length $D$, namely,

$$q^{th} \text{ segment kept if: } 100\frac{t_{q,2} - t_{q,1}}{D} \geq C, \tag{64}$$

where $t_{q,1}$ and $t_{q,2}$ are the times of the first and last data points inside in the $q^{th}$ segment, and $C$ is the coverage factor. Its default value in WAVEPAL is 90 %.

Third, the frequency range on the $q^{th}$ segment is bounded by these two frequencies:

$$f_{\min} = \frac{1}{t_{q,2} - t_{q,1}} \quad \text{and} \quad f_{\max} = \frac{1}{2\overline{\Delta t_q}}. \tag{65}$$



The maximal period $(1/f_{\min})$ corresponds to the effective length on the segment. The maximal frequency in the case of regularly sampled data must be the Nyquist frequency, $f_{\max} = 1/2\Delta t$. For irregularly sampled data, different choices for $\overline{\Delta t}_q$ are possible. As suggested in appendix A, an option is $\overline{\Delta t}_q = \Delta t_{\mathrm{GCD,q}}$, but this choice is insufficient to avoid *pseudo-aliasing* issues. Imagine for example a regularly sampled time series with 1000 data points and $\Delta t = 1$.

Add one point at the end with the last time step being 0.1. The resulting irregularly sampled time series will thus have $\Delta t_{\mathrm{GCD}} = 0.1$. If we take $f_{\max} = 5$, it is obvious that some kind of aliasing will occur between $f = 0.5$ and $f_{\max}$. This it what we call *pseudo-aliasing*. A much better choice in this case is of course $f_{\max} = 0.5$. (Bretthorst, 2001, Sect. 5) provides further discussions on this topic.

In practice,

$$\overline{\Delta t}_q = \max\left\{ \frac{\sum_{k=1}^{N} G_{q_{k,k}}\Delta t_{c_k}}{\mathrm{tr}(G_q)}, \frac{\sum_{k=1}^{N-1} H_{q_{k,k}}\Delta t_k}{\mathrm{tr}(H_q)} \right\}, \tag{66}$$

where

$$\Delta t_k = t_{k+1} - t_k, \; \Delta t_{c_k} = \frac{t_{k+1} - t_{k-1}}{2} \; \forall k \in \{2, ... N-1\}, \; \Delta t_{c_1} = t_2 - t_1, \; \Delta t_{c_N} = t_N - t_{N-1}, \tag{67}$$

and

$$H_q = \text{diagonal matrix with } H_{q_{k,k}} = \text{taper at time } \frac{t_k + t_{k+1}}{2}, \; k \in \{1, ..., N-1\}. \tag{68}$$

appears to work well. More justification and an example will be provided in part II of this study (Lenoir and Crucifix, 2017). Matrix $H_q$ is similar to matrix $G_q$, defined in Sect. 4.4, but with elements taken at $(t_k + t_{k+1})/2$ instead of $t_k$. Quantity $\overline{\Delta t}_q$ is equal to the maximum between the average time step and the average central time step if there is no tapering ($G_q = H_q = \mathbb{I}$), and is equal to $\Delta t$ in the regularly sampled case. These restrictions on the frequency bounds imply that the total number of WOSA segments, $Q$, in formula (58), is not the same for all the frequencies. This is

illustrated on Fig. 8b.

Fourth, in order the have a reliable average of the periodograms, we only represent the periodogram at the frequencies for which the number of WOSA segments is above some threshold. In WAVEPAL, default value for the threshold at frequency $f$ is

$$\text{Threshold: } \min\{10, \max_{\{f\}} Q(f)\}, \tag{69}$$

where $Q(f)$ is the number of WOSA segments at frequency $f$. It means that frequency $f$ belongs to the range of frequencies of the WOSA periodogram if $Q(f)$ is greater than or equal to the threshold.

## 5  Significance testing with the periodogram

### 5.1  Hypothesis testing

Significance testing allows us to test for the presence of periodic components in the signal. It is mathematically

expressed as a hypothesis testing, see (Brockwell and Davis, 1991, Chap. 10). Taking our model, Eq. (12), the null



hypothesis is

$$H_0 : A_\omega = B_\omega = 0. \tag{70}$$

Therefore, $|X\rangle = |\text{Trend}\rangle + |\text{Noise}\rangle$. The alternative hypothesis is

$$H_1 : A_\omega \text{ and } B_\omega \text{ are not both zero.} \tag{71}$$

The decision of accepting or rejecting the null hypothesis is based on the periodogram (formula (58)), independently for each frequency (*pointwise testing*). Concretely, for each frequency, we compute the distribution of the periodogram under the null hypothesis, and then see if the *data* periodogram at that frequency is above or below a given percentile (e.g. the 95$^{\text{th}}$) of that distribution. The percentile is called *level of confidence*. If the data periodogram is above the $X^{\text{th}}$ percentile of the reference distribution, we reject the null hypothesis with $X$ % of confidence. The *level of significance* is equal to $(100 - X)$ %, e.g. a 95 % confidence level is equivalent to a 5 % significance level. See Fig. 8c and 8d for an illustration. We recommend (Priestley, 1981, Chap. 6) for more details on the methodology.

To perform significance testing, we thus need

1. to estimate the parameters of the process under the null hypothesis. This is studied in Sect. 5.2.

2. to estimate the distribution of the periodogram under the null hypothesis. This is studied in Sect. 5.3.

## 5.2 Estimation of the parameters under the null hypothesis

### 5.2.1 Introduction

Under the null hypothesis, the signal is $|X\rangle = |\text{Trend}\rangle + |\text{Noise}\rangle$, and we thus need to estimate the parameters of the trend and those of the zero-mean CARMA process. The best statistical approach is to estimate them jointly, and marginalize over the parameters of the trend, since the periodogram is invariant with respect to these parameters, according to Sect. 4.3. But this would imply very involved computations that are way beyond the scope of this work. We are thus forced to a compromise and proceed as follows: data are detrended, $|X_{\text{det}}\rangle = |X\rangle - P_{\overline{\text{sp}}\{\mathbf{t^0}, \mathbf{t^1}, \dots, \mathbf{t^m}\}} |X\rangle$, and then we estimate the parameters of the CARMA process, based on the model $\mu|t^0\rangle + |\text{Noise}\rangle$, where $|\text{Noise}\rangle$ is a zero-mean stationary Gaussian CARMA process sampled at the times of $|t\rangle$.

Estimation of CARMA parameters is done in a Bayesian framework. We analyze separately the case of the white noise, which is done analytically, and the case of CARMA(p,q) processes with $p \geq 1$, for which Markov-Chain Monte-Carlo (MCMC) methods are required. Bayesian analysis provides a posterior distribution of the parameters based on priors.





### 5.2.2   Gaussian white noise

We want to estimate the two parameters of the white noise, the mean $\mu$ and the variance $\sigma^2$. According to the Bayes theorem:

$$\Pi(\mu, \sigma^2|D) = \frac{\Pi(D|\mu, \sigma^2)\Pi(\mu, \sigma^2)}{\Pi(D)} \sim \Pi(D|\mu, \sigma^2)\Pi(\mu, \sigma^2), \tag{72}$$

where $\Pi$ is the probability density function (PDF) and D is the detrended data $X_{\text{det},1}, ..., X_{\text{det},N}$. Based on the PDF of a multivariate white noise, the likelihood function is

$$\Pi(D|\mu, \sigma^2) = \left(\sqrt{\frac{1}{2\pi\sigma^2}}\right)^N \exp\left(\frac{-\sum_{i=1}^{N}(X_{\text{det},i} - \mu)^2}{2\sigma^2}\right). \tag{73}$$

We take *Jeffreys priors* (Jeffreys, 1946) for $\mu$ and $\sigma^2$:

$$\Pi(\mu, \sigma^2) = \Pi(\mu)\Pi(\sigma^2), \text{ with } \Pi(\mu) \sim 1 \text{ and } \Pi(\sigma^2) \sim \frac{1}{\sigma^2} \tag{74}$$

Jeffreys priors are non-informative and invariant under reparametrization. Note that $\Pi(\sigma^2)$ is log-uniform.

Since we do not actually need to estimate $\mu$ (see Sect. 4.3 and formula (58)), we marginalize over that variable,

$$\Pi(\sigma^2|D) = \int\limits_{-\infty}^{+\infty} \mathrm{d}\mu\, \Pi(\mu, \sigma^2|D)$$

$$\sim \frac{1}{\sigma^2} \int\limits_{-\infty}^{+\infty} \mathrm{d}\mu\, \Pi(D|\mu, \sigma^2)$$

$$\sim \frac{1}{\sigma^2} \left(\sqrt{\frac{1}{2\pi\sigma^2}}\right)^N \exp\left(\frac{-\sum_{i=1}^{N}X_{\text{det},i}^2}{2\sigma^2}\right) \int\limits_{-\infty}^{+\infty} \mathrm{d}\mu\, \exp(-(a\mu^2 + 2b\mu))$$

$$\sim \frac{1}{\sigma^2} \left(\sqrt{\frac{1}{2\pi\sigma^2}}\right)^N \exp\left(\frac{-\sum_{i=1}^{N}X_{\text{det},i}^2}{2\sigma^2}\right) \sqrt{\frac{\pi}{a}} \exp\left(\frac{b^2}{a}\right), \tag{75}$$

with $a = N/2\sigma^2$ and $b = -\sum_{i=1}^{N}X_{\text{det},i}/2\sigma^2$. Rearranging terms gives

$$\Pi(\sigma^2|D) \sim \left(\frac{1}{\sigma^2}\right)^{\frac{N+1}{2}} \exp\left(-\frac{1}{\beta\sigma^2}\right), \tag{76}$$

with $\beta = 2/N\hat{\sigma}^2$, where $\hat{\sigma}^2 =$ is the (biased) variance of the detrended data[5]. With the variable change $y = 1/\sigma^2$, we have

$$\Pi(y|D) \sim y^{\frac{N-3}{2}} \exp(-y/\beta), \tag{77}$$

which is nothing but a gamma distribution:

$$\frac{1}{\sigma^2} \stackrel{d}{=} \gamma\left(\frac{N-1}{2}, \frac{2}{N\hat{\sigma}^2}\right). \tag{78}$$

---

[5] $\hat{\sigma}^2 = \frac{1}{N}\sum_{i=1}^{N}X_{\text{det},i}^2 - \left(\frac{1}{N}\sum_{i=1}^{N}X_{\text{det},i}\right)^2$





Note that the mean of the distribution in Eq. (78) is equal to $(N-1)/(N\hat{\sigma}^2)$, which is the usual unbiased estimator of $1/\sigma^2$. Finally, the PDF of $\sigma^2$ is maximum at

$$\sigma_{\max}^2 = \frac{N}{N+1}\hat{\sigma}^2. \tag{79}$$

This is obtained from the derivative of Eq. (76).

### 5.2.3 Gaussian CARMA$(p,q)$ noise with $p \geq 1$

For other cases than the white noise, (Kelly et al., 2014) provide robust algorithms to estimate the posterior distribution of the CARMA parameters and of the parameter $\mu$ of an irregularly sampled, purely stochastic, time series, which can be modeled as a CARMA process. Those algorithms are based on Bayesian inference and MCMC methods. We recommend to read in particular Sect. 3.3 and 3.6 of (Kelly et al., 2014) for a discussion on the choice of the priors and for computational considerations respectively. That paper is accompanied by a Python and C++ package called *CARMA pack*. Some outputs of CARMA pack are shown in Sect. 9.

## 5.3 Estimation of the distribution of the periodogram under the null hypothesis

### 5.3.1 Working with a trendless stochastic process

Under the null hypothesis, the signal is $|X\rangle = |\text{Trend}\rangle + |\text{Noise}\rangle = \sum_{k=0}^{m}\gamma_k|t^k\rangle + |\text{Noise}\rangle$. The WOSA periodogram, Eq. (58), being invariant with respect to the parameters of the trend, we can pose $\gamma_k = 0$ for all $k$ and $|X\rangle$ reduces to a zero-mean CARMA process.

### 5.3.2 Monte-Carlo approach

For each frequency, we need the distribution of the WOSA periodogram, Eq. (62), where $|X\rangle$ is now a CARMA process for which we know the distribution of its parameters, from Sect. 5.2. With Monte-Carlo methods, we are thus able to estimate any percentile of the distribution of the periodogram. If $|X\rangle$ is a zero-mean white noise, $|X\rangle$ is sampled from a standard normal distribution multiplied by the square root of the variance, whose inverse is sampled from the gamma distribution (Eq. (78)). If $|X\rangle$ is a CARMA$(p,q)$ process with $p \geq 1$, $|X\rangle$ is generated with the Kalman filter (from CARMA pack - see Sect. 5.2.3). An example of confidence levels is shown on Fig. 8d.

We are thus able to estimate confidence levels for the WOSA periodogram taking into account the uncertainty on the parameters of the background noise.

### 5.3.3 Analytical approach

If we consider constant CARMA parameters, we show in this Sect. that analytical confidence levels can be computed, even in the very tail of the distribution of the periodogram of the background noise. An example is given on Fig. 8c. The advantage of the analytical approach is double:



1. It provides confidence levels converging to the exact solution, as the number of conserved moments increases (see below). From a certain number of conserved moments, we can consider that convergence is numerically reached (see Fig. 9). Such an approach is particularly interesting for high confidence levels, as illustrated on Fig. 8c with the 99.9 % confidence level, for which a MCMC approach would require a huge number of samples to get a satisfactory accuracy.

2. As a consequence, for a given percentile, computing time is usually shorter with the analytical method than with the MCMC method. We note, however, that the MCMC approach generally needs less computing time when the number of data points becomes large, as shown in appendix E.

#### 5.3.4   First approximation:

If the marginal posterior distribution of each CARMA parameter is unimodal, we take the parameter value at the maximum of its PDF (white noise case, see Eq. (79)), or the median parameter[6] (other cases). Note that multimodality tends to appear more frequently for CARMA processes of high order. Working with a unique set of parameters allows us to find an analytical formula for the distribution of the WOSA periodogram. Considering the matrix forms of the CARMA noise (Eq. (15) or (34)) and the WOSA periodogram (Eq. (62)), we demonstrate the following theorem.

**Theorem 1.** *The WOSA periodogram, defined in Eq. (62), under the null hypothesis (70), is*

$$||P_{WOSA}(\omega)|X\rangle||^2 \overset{d}{=} \sum_{k=1}^{2Q(\omega)} \lambda_k(\omega)\chi_1^2, \tag{80}$$

*where $|X\rangle = \sum_{k=0}^{m} \gamma_k |t^k\rangle + K|Z\rangle$, $K$ is the CARMA matrix defined in Eq. (15) or (34), and $Q(\omega)$ is the number of WOSA segments at $\omega$.*

*The $\chi_1^2$ distributions are iid, and $\lambda_1(\omega)$, ..., $\lambda_{2Q(\omega)}(\omega)$ are the eigenvalues of $M'_{2,\omega}KK'M_{2,\omega}$ and are non-negative. Matrix $M_{2,\omega}$ is defined in Eq. (63).*

*Proof.* Since the WOSA periodogram, Eq. (62), is invariant with respect to the parameters of the trend, we pose them equal to zero and consider the zero-mean CARMA process

$$|X\rangle = K|Z\rangle. \tag{81}$$

The periodogram is thus

$$||P_{\text{WOSA}}(\omega)|X\rangle||^2 = \langle Z|K'M_{2,\omega}M'_{2,\omega}K|Z\rangle = \gamma'\gamma, \tag{82}$$

---

[6]For CARMA processes with $p > 0$ and $q \geq 0$, the marginal posterior distribution is obtained by MCMC methods, and determining the maximum of the PDF thus requires some post-processing, such as smoothing the distribution. An alternative is to simply take the median.



with $\gamma = M'_{2,\omega}K|Z\rangle$. Since $|Z\rangle$ is a standard multivariate normal distribution, we have

$$\gamma \overset{d}{=} \mathcal{N}(0, M'_{2,\omega}KK'M_{2,\omega}). \tag{83}$$

$M'_{2,\omega}KK'M_{2,\omega}$ is a $(2Q(\omega), 2Q(\omega))$ real symmetric positive semi-definite matrix. We can thus diagonalize it:

$$\exists \text{ an orthogonal matrix } U \text{ s.t. } U'M'_{2,\omega}KK'M_{2,\omega}U = D, \tag{84}$$

with $D$ being a diagonal matrix with the $2Q(\omega)$ non-negative eigenvalues of $M'_{2,\omega}KK'M_{2,\omega}$. We now have $U'\gamma \overset{d}{=} \mathcal{N}(0, D)$, and

$$||P_{\text{WOSA}}(\omega)|X\rangle||^2 = \gamma'\gamma = \gamma'UU'\gamma = \langle Z|\sqrt{D}\sqrt{D}|Z\rangle \overset{d}{=} \sum_{k=1}^{2Q(\omega)} \lambda_k(\omega)\chi_1^2, \tag{85}$$

where the $\chi_1^2$ distributions are iid.                                                                                      □

### 5.3.5   Pseudo-spectrum

The *pseudo-spectrum* is defined as the expected value of the periodogram distribution:

$$\widehat{S}(\omega) = \sum_{k=1}^{2Q(\omega)} \lambda_k(\omega) = \text{tr}(M'_{2,\omega}KK'M_{2,\omega}). \tag{86}$$

The difference between the *pseudo-spectrum* and the traditional *spectrum* is explained in appendix C.

### 5.3.6   White noise

If the background noise is white, we have $K = \sigma\mathbb{I}$ and this implies that $\text{tr}(M'_{2,\omega}KK'M_{2,\omega}) = \text{tr}(M'_{2,\omega}M_{2,\omega})\sigma^2 =$
$\text{tr}(M_{2,\omega}M'_{2,\omega})\sigma^2 = 2\sigma^2$, such that the pseudo-spectrum is

$$\widehat{S}(\omega) = 2\sigma^2, \tag{87}$$

and is thus flat. This is a well-known result of the LS periodogram (Scargle, 1982), generalized here to more evolved periodograms. Moreover, if there is no WOSA segmentation ($Q(\omega) = 1 \; \forall\omega$), the periodogram is exactly chi-square distributed with 2 degrees of freedom:

$$||(P_{\overline{\text{sp}}\{\mathbf{t^0},\mathbf{t^1},...,\mathbf{t^m},\mathbf{Gc}_\omega,\mathbf{Gs}_\omega\}} - P_{\overline{\text{sp}}\{\mathbf{t^0},\mathbf{t^1},...,\mathbf{t^m}\}})\sigma|Z\rangle||^2 \overset{d}{=} \sigma^2\chi_1^2 + \sigma^2\chi_1^2 \overset{d}{=} \sigma^2\chi_2^2, \tag{88}$$

which is also a generalization of a well-known result of the LS periodogram (Scargle, 1982).

### 5.3.7   Variance

The variance of the distribution of the periodogram, Eq. (80), is equal to $2\sum_{k=1}^{2Q(\omega)} \lambda_k^2(\omega) = 2||M'_{2,\omega}KK'M_{2,\omega}||_F^2$, where $||\cdot||_F$ is the Frobenius norm. As expected, it decreases with Q, as illustrated on Fig. 3.




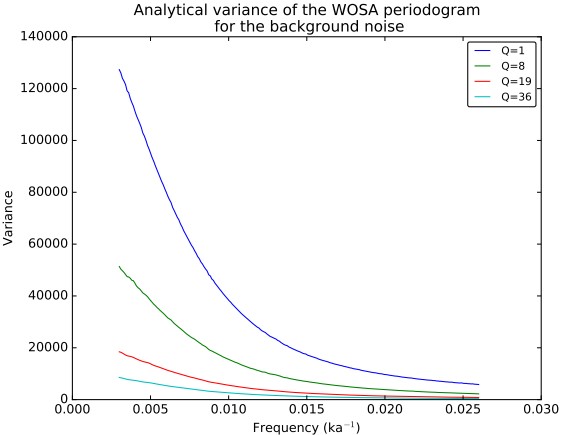

**Figure 3.** Analytical variance of the WOSA periodogram for a Gaussian red noise with $\sigma = 2$ and $\alpha = 1/20$ (see Sect. 3.2.3 for the definition of a red noise) for different values of $Q$. The frequency range is chosen such that, for each curve, $Q(\omega)$ is constant all along. The red noise is built on the irregularly sampled times of ODP1148 core (see Sect. 9).

Going back to Eq. (80), it is well-known that a linear combination of (independent) $\chi^2$ distributions is not analytically solvable. Fortunately, excellent approximations are available in (Provost et al., 2009), allowing to avoid Monte-Carlo methods.

5  **5.3.8  Second approximation:**

We approximate the linear combination of independent chi-square distributions, conserving its first $d$ moments. When $d \to \infty$, the approximation converges to the exact distribution. In practice, estimation of a percentile is already very good with a very few moments, as illustrated on Fig. 9. Let us proceed step by step by increasing the number of conserved moments. Define $X = \sum_{k=1}^{2Q(\omega)} \lambda_k(\omega)\chi_1^2$.

10  **5.3.9  1-moment approximation**

We require the expected value of the process to be conserved, which is satisfied with the following approximation:

$$X \stackrel{d}{\approx} \frac{1}{2Q(\omega)}\Big[\sum_{k=1}^{2Q(\omega)} \lambda_k(\omega)\Big]\chi_{2Q(\omega)}^2, \tag{89}$$

or, equivalently,

$$X \stackrel{d}{\approx} \frac{1}{2Q(\omega)}\widehat{S}(\omega)\chi_{2Q(\omega)}^2. \tag{90}$$





### 5.3.10 2-moment approximation

The approximate distribution of the linear combination of the chi-square distributions must have two parameters, and we conserve the expected value and variance. A chi-square distribution with $M$ degrees of freedom provides a good fit:

$$X \stackrel{d}{\approx} g\chi^2_M. \tag{91}$$

Equating the expected values and variances gives

$$M = \frac{(\mathrm{tr}(A))^2}{||A||^2_F} \text{ and } g = \frac{||A||^2_F}{\mathrm{tr}(A)}, \tag{92}$$

where $A = M'_{2,\omega} K K' M_{2,\omega}$ and $||A||^2_F$ is the squared Frobenius norm of matrix $A$, i.e. the sum of its squared eigenvalues. Note that $g\chi^2_M \stackrel{d}{=} \gamma_{M/2,2g}$, where $2g$ is the *scale* parameter of the gamma distribution, which motivates the following d-moment approximation.

### 5.3.11 d-moment approximation

We apply here the formulas presented in (Provost et al., 2009). Let $f_X$ be the PDF of $X$. This distribution is approximated by the PDF of a $d^{\mathrm{th}}$ degree gamma-polynomial distribution:

$$f_X(x) \approx \gamma_{\alpha,\beta}(x) \sum_{i=0}^{d} \xi_i x^i, \qquad x \geq 0, \tag{93}$$

where the parameters $\alpha$ and $\beta$ are estimated with the 2-moment approximation detailed above. $\xi_0, ..., \xi_d$ are the solution of

$$\begin{pmatrix} \xi_0 \\ \xi_1 \\ \vdots \\ \xi_d \end{pmatrix} = \begin{pmatrix} \eta(0) & \eta(1) & \dots & \eta(d-1) & \eta(d) \\ \eta(1) & \eta(2) & \dots & \eta(d) & \eta(d+1) \\ \vdots & \vdots & \vdots & \vdots & \vdots \\ \eta(d) & \eta(d+1) & \dots & \eta(2d-1) & \eta(2d) \end{pmatrix}^{-1} \begin{pmatrix} 1 \\ \mu(1) \\ \vdots \\ \mu(d) \end{pmatrix}. \tag{94}$$

$\mu(1), ..., \mu(d)$ are the exact first $d$ moments of $X$ and can be computed analytically by recurrence (Provost et al., 2009, Eq. (5)). $\eta(h)$ is the $h^{\mathrm{th}}$ moment of the gamma distribution, $\eta(h) = \beta^h \Gamma(\alpha+h)/\Gamma(\alpha)$. The approximate cumulative distribution function (CDF) of $X$, evaluated at $c_0$, is then

$$F_X(c_0) \approx \frac{1}{\Gamma(\alpha)} \sum_{i=0}^{d} \xi_i \beta^i \gamma(i+\alpha, c_0/\beta), \qquad c_0 > 0, \tag{95}$$

where $\gamma(s,x)$ is the lower incomplete gamma function:

$$\gamma(s,x) = \int_0^x \mathrm{d}t \; t^{s-1} \exp(-t). \tag{96}$$




After all that chain of calculus, we reached our objective, that is, the estimation of a confidence level for the WOSA periodogram. It is given by the solution $c_0$ of

$$\frac{1}{\Gamma(\alpha)}\sum_{i=0}^{d}\xi_i\beta^i\gamma(i+\alpha,c_0/\beta)-p=0, \tag{97}$$

for some p-value $p$, e.g. $p=0.95$ for a 95 % confidence level.

The gamma-polynomial approximation can be extended to the *generalized* gamma-polynomial approximation. The latter is based on the generalized gamma distribution and is defined in appendix D. It gives percentiles that usually converge faster than with the gamma-polynomial approximation. However, we observed that the generalized gamma-polynomial approximation is quite sensitive to the quality of the first guess for the three parameters of the generalized gamma distribution (see appendix D). We thus recommend the use of the gamma-polynomial approximation as a

first choice. Both options are available in WAVEPAL.

Finally, we mention that there exists an alternative expression to the above development, in terms of Laguerre polynomials, see (Provost, 2005). It has the advantage of not requiring the matrix inversion in Eq. (94), the latter possibly being singular at large values of the degree $d$. However, we have not found any improvement on the stability or computing time using that approach.

**5.4    The F-periodogram for the white noise background**

We shown in Eq. (88) that the periodogram of a Gaussian white noise is exactly chi-square distributed if there is no WOSA segmentation. Significance testing against a white noise requires the estimation of the white noise variance after having detrended the data. Knowing that a F-distribution is the ratio of independent chi-square distributions, it is possible to get rid of the detrending and variance estimation and deal with a well-known distribution, by working

with

$$\frac{(N-m-3)||(P_{\overline{\mathrm{sp}}\{\mathbf{t^0},\mathbf{t^1},...,\mathbf{t^m},\mathbf{c}_\omega,\mathbf{s}_\omega\}}-P_{\overline{\mathrm{sp}}\{\mathbf{t^0},\mathbf{t^1},...,\mathbf{t^m}\}})|X\rangle||^2}{2||(\mathbb{I}-P_{\overline{\mathrm{sp}}\{\mathbf{t^0},\mathbf{t^1},...,\mathbf{t^m},\mathbf{c}_\omega,\mathbf{s}_\omega\}})|X\rangle||^2}. \tag{98}$$

We call it the *F-periodogram*. We already know that the numerator is invariant with respect to the parameters of the trend of the signal. It is clear that the denominator is invariant with respect to the parameters of the trend as well as with respect to the amplitudes of the periodic components (only the $|\mathrm{Noise}\rangle$ term remains when applying it to Eq.

(12)). Moreover, that ratio is invariant with respect to the variance of the signal. Last but not least, the orthogonal projections in the numerator, $[P_{\overline{\mathrm{sp}}\{\mathbf{t^0},\mathbf{t^2},...,\mathbf{t^m},\mathbf{c}_\omega,\mathbf{s}_\omega\}}-P_{\overline{\mathrm{sp}}\{\mathbf{t^0},\mathbf{t^2},...,\mathbf{t^m}\}}]$, and in the denominator, $[\mathbb{I}-P_{\overline{\mathrm{sp}}\{\mathbf{t^0},\mathbf{t^2},...,\mathbf{t^m},\mathbf{c}_\omega,\mathbf{s}_\omega\}}]$, are done on spaces that are orthogonal to each other. Consequently, if we consider the null hypothesis (70) with a white noise, the numerator and the denominator follow **independent** chi-square distributions, and

$$\frac{(N-m-3)||(P_{\overline{\mathrm{sp}}\{\mathbf{t^0},\mathbf{t^1},...,\mathbf{t^m},\mathbf{c}_\omega,\mathbf{s}_\omega\}}-P_{\overline{\mathrm{sp}}\{\mathbf{t^0},\mathbf{t^1},...,\mathbf{t^m}\}})|X\rangle||^2}{2||(\mathbb{I}-P_{\overline{\mathrm{sp}}\{\mathbf{t^0},\mathbf{t^1},...,\mathbf{t^m},\mathbf{c}_\omega,\mathbf{s}_\omega\}})|X\rangle||^2} \overset{d}{=} \frac{(N-m-3)\chi_2^2}{2\chi_{N-m-3}^2}$$

$$\overset{d}{=} F(2,N-m-3), \tag{99}$$




where $|X\rangle \overset{d}{=} \sum_{k=0}^{m} \gamma_k |t^k\rangle + \mathcal{N}(\mu, \sigma^2) \overset{d}{=} |\text{Trend}\rangle + \mathcal{N}(\mu, \sigma^2),$     (100)

and where $F(2, N - m - 3)$ is the Fisher-Snedecor distribution with parameters 2 and $N - m - 3$. In conclusion, the F-periodogram can be an alternative to the periodogram when performing significance testing. It has the advantage of not requiring any parameter to be estimated and applies under the following conditions

- The background noise is assumed to be white

- There is no WOSA segmentation

- There is no tapering

The F-periodogram is available in WAVEPAL under the above requirements.

With a WOSA segmentation, projections at the numerator and at the denominator are not performed anymore on orthogonal spaces, and this cannot therefore be applied.

The above results are a generalization of formulas in (Brockwell and Davis, 1991) and (Heck et al., 1985). See appendix F for additional details.

## 6 The amplitude periodogram

### 6.1 Definition

Going back to Eq. (12), we now look for the amplitude $E_\omega = \sqrt{A_\omega^2 + B_\omega^2}$ at a given frequency $f = \frac{\omega}{2\pi}$. The estimation of $E_\omega^2$ is called the *amplitude periodogram* and is denoted by $\widehat{E}_\omega^2$. We estimate $A_\omega$ and $B_\omega$ with a least squares approach. We start with a trendless signal, and will show that the amplitude periodogram and the periodogram are approximately proportional.

### 6.2 Trendless signal

#### 6.2.1 No tapering

The estimated amplitudes we look for, $\widehat{A_\omega}$ and $\widehat{B_\omega}$, are the solution of

$$(\widehat{A}_\omega, \widehat{B}_\omega) = \underset{\{(A,B)\in\mathbb{R}^2\}}{\text{argmin}} \; || \, |X\rangle - (A|c_\omega\rangle + B|s_\omega\rangle) ) ||^2.$$     (101)

Since we look for the minimal distance, the solution is given by the orthogonal projection onto the vector space spanned by $|c_\omega\rangle$ and $|s_\omega\rangle$, namely

$$P_{\overline{\text{sp}}\{\mathbf{c}_\omega, \mathbf{s}_\omega\}} |X\rangle = \widehat{A}_\omega |c_\omega\rangle + \widehat{B}_\omega |s_\omega\rangle.$$     (102)





Let us develop this equation:

$$V_2(V_2'V_2)^{-1}V_2'|X\rangle = V_2|\widehat{\Phi}_\omega\rangle, \tag{103}$$

where

$$V_2 = \begin{pmatrix} | & | \\ |c_\omega\rangle & |s_\omega\rangle \\ | & | \end{pmatrix} \text{ and } |\widehat{\Phi}_\omega\rangle = \begin{pmatrix} \widehat{A}_\omega \\ \widehat{B}_\omega \end{pmatrix}, \tag{104}$$

and we find the well-known expression for the solution of a least squares problem:

$$|\widehat{\Phi}_\omega\rangle = (V_2'V_2)^{-1}V_2'|X\rangle. \tag{105}$$

Finally,

$$\widehat{E}_\omega = |||\widehat{\Phi}_\omega\rangle||. \tag{106}$$

In the regularly sampled case, at the Fourier frequencies, the amplitude periodogram is proportional to the periodogram, with a factor $2/N$ (or a factor $1/N$ at $\omega = 0$ and $\pi/\Delta t$ ; the projection being done on the single cosine at those frequencies). It is not anymore the case with irregularly sampled time series, and the proportionality is only approximate:

$$\widehat{E}_\omega^2 \approx \frac{2}{N}||P_{\overline{\mathrm{sp}}\{\mathbf{c}_\omega, \mathbf{s}_\omega\}}|X\rangle||^2. \tag{107}$$

To prove the above formula, rewrite the model (12):

$$|X\rangle = E_\omega \cos(\omega|t\rangle + \phi_\omega - \beta_\omega + \beta_\omega) + |\mathrm{Noise}\rangle$$
$$= A_\omega \cos(\omega|t\rangle - \beta_\omega) + B_\omega \sin(\omega|t\rangle - \beta_\omega) + |\mathrm{Noise}\rangle, \tag{108}$$

where $\beta_\omega$ is defined in Eq. (38) and makes the phase-lagged sine and cosine orthogonal. $A_\omega$ and $B_\omega$ no longer have the same expressions as in Eq. (12), but we still have $E_\omega^2 = A_\omega^2 + B_\omega^2$. We can rewrite Eq. (105) but this time with $V_2$ holding the above phase-lagged sine and cosine. We now make use of the approximation stated in (Lomb, 1976, p. 449):

$$\sum_{i=1}^{N} \cos^2(\omega t_i - \beta_\omega) \approx \frac{N}{2} \quad \text{and} \quad \sum_{i=1}^{N} \sin^2(\omega t_i - \beta_\omega) \approx \frac{N}{2}. \tag{109}$$

Note that the sum of both is exactly equal to $N$. Equation (107) is then obtained observing that $V_2'V_2 \approx \frac{N}{2}\mathbb{I}$. Basic trigonometry gives the following equalities for the relative error of the above approximations:

$$\left|\frac{\sum_{i=1}^{N} \cos^2(\omega t_i - \beta_\omega) - N/2}{N/2}\right| = \left|\frac{\sum_{i=1}^{N} \sin^2(\omega t_i - \beta_\omega) - N/2}{N/2}\right| = \left|\frac{\sum_{i=1}^{N} \cos(2(\omega t_i - \beta_\omega))}{N}\right|, \tag{110}$$

so that the two approximations of (109) reduce to only one:

$$\frac{\sum_{i=1}^{N} \cos(2(\omega t_i - \beta_\omega))}{N} \approx 0. \tag{111}$$

The quality of this approximation is illustrated on Fig. 4.





### 6.2.2 With tapering

Like with the periodogram, leakage also appears in the amplitude periodogram. Consequently, it may be better to work with the projection on tapered cosine and sine if the data are not too much irregularly sampled, as explained in Sect. 4.4. Considering the tapered case is also an important mathematical prerequisite for an extension to the
continuous wavelet transform. This is developed in part II of this study (Lenoir and Crucifix, 2017).

$\widehat{A}_\omega$ and $\widehat{B}_\omega$ are determined by projecting the data onto tapered cosine and sine:

$$P_{\overline{\mathrm{sp}}\{\mathbf{Gc}_\omega,\mathbf{Gs}_\omega\}}|X\rangle = \widehat{A}_\omega|c_\omega\rangle + \widehat{B}_\omega|s_\omega\rangle. \tag{112}$$

Developing the equation gives

$$|\widehat{\Phi}_\omega\rangle = (V_2'GV_2)^{-1}V_2'G|X\rangle, \tag{113}$$

and

$$\widehat{E}_\omega = |||\widehat{\Phi}_\omega\rangle||, \tag{114}$$

where $V_2$ is defined in Sect. 6.2.1 and $G$ is defined in Sect. 4.4.

Note that the approach we follow does not correspond to the classical least squares problem as above since, in Eq. (112), the cosine and sine are tapered only on the left-hand side of the equality. However, one can reconstruct a sig-
nal from its projection coefficients with another function than the one which is used to determine those coefficients. See (Torrésani, 1995, Eq. (II.8) p. 15) in which the similarity with $V_2|\widehat{\Phi}_\omega\rangle = V_2(V_2'GV_2)^{-1}V_2'G|X\rangle$ is evident. Note that $V_2(V_2'GV_2)^{-1}V_2'G$ is a projection, since it is idempotent, but the projection is not orthogonal, because it is not symmetric.

Similarly to the non-tapered case, we now determine an approximate proportionality between the amplitude peri-
odogram and the tapered periodogram. We start with the model (12) which is written under the following form

$$|X\rangle = A_\omega \cos(\omega|t\rangle - \beta_\omega) + B_\omega \sin(\omega|t\rangle - \beta_\omega) + |\mathrm{Noise}\rangle, \tag{115}$$

where $\beta_\omega$ is introduced such that $\langle Gc_\omega | Gs_\omega \rangle = 0$, or equivalently, such that $V_2'G^2V_2$ is diagonal. A little development gives the formula for determining $\beta_\omega$:

$$\tan(2\beta_\omega) = \frac{\sum_{i=1}^N G_{ii}^2 \sin(2\omega t_i)}{\sum_{i=1}^N G_{ii}^2 \cos(2\omega t_i)}, \tag{116}$$

which is a generalization of Eq. (38). We now make use of the following approximations:

$$\frac{\sum_{i=1}^N G_{ii} \cos(2(\omega t_i - \beta_\omega))}{\mathrm{tr}(G)} \approx 0, \tag{117a}$$

$$\frac{\sum_{i=1}^N G_{ii}^2 \cos(2(\omega t_i - \beta_\omega))}{\mathrm{tr}(G^2)} \approx 0, \tag{117b}$$



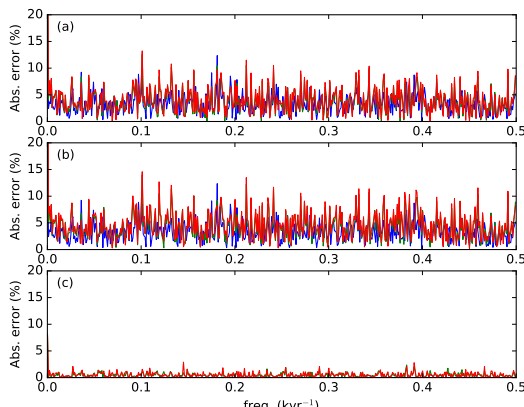

**Figure 4.** Illustration of the quality of the approximations (a) Eq. (117a) (b) Eq. (117b) and (c) Eq. (119). In blue: No tapering (square taper), in green: $\sin^2$ taper, in red: Gaussian taper. The approximation (111) is thus in blue in (a) or (b). Each panel represents the left-hand side of the equation, multiplied by 100, to express percentage. This indicates how small is the numerator compared to the denominator. The time vector $|t\rangle$ comes from the ODP1148 core (see Sect. 9) for which $\Delta t_{\text{GCD}} = 1$ kyr.

which are similar to the approximation made in (111). That implies, with no extra approximation, the following formulas:

$$\sum_{i=1}^{N} G_{ii} \cos^2(\omega t_i - \beta_\omega) \approx \frac{\text{tr}(G)}{2}, \qquad \sum_{i=1}^{N} G_{ii} \sin^2(\omega t_i - \beta_\omega) \approx \frac{\text{tr}(G)}{2}, \tag{118a}$$

$$\sum_{i=1}^{N} G_{ii}^2 \cos^2(\omega t_i - \beta_\omega) \approx \frac{\text{tr}(G^2)}{2}, \quad \text{and} \quad \sum_{i=1}^{N} G_{ii}^2 \sin^2(\omega t_i - \beta_\omega) \approx \frac{\text{tr}(G^2)}{2}. \tag{118b}$$

5 Note that in (118a) and (118b), the sum of the two members is conserved and we find back Eq. (109) when $G = \mathbb{I}$. Moreover, we approximate the following sum:

$$\frac{\sum_{i=1}^{N} G_{ii} \cos(\omega t_i - \beta_\omega) \sin(\omega t_i - \beta_\omega)}{\text{tr}(G)/2} \approx 0, \tag{119}$$

so that $V_2' G V_2$ is diagonal. The quality of these approximations is illustrated on Fig. 4. Putting all together gives

$$V_2' G V_2 \approx \frac{\text{tr}(G)}{2} \mathbb{I}, \quad \text{and} \quad V_2' G^2 V_2 \approx \frac{\text{tr}(G^2)}{2} \mathbb{I}, \tag{120}$$

10 from which we deduce

$$\widehat{E}_\omega^2 \approx \frac{2 \text{tr}(G^2)}{\text{tr}(G)^2} || P_{\overline{\text{sp}}\{\mathbf{Gc}_\omega, \mathbf{Gs}_\omega\}} |X\rangle ||^2. \tag{121}$$

Finally, we mention that the above relation is approximate as well in the case of regularly sampled time series.





## 6.3 Signal with a trend

We now work with the full model (12) including the trend. Our aim is again to find the amplitude $E_\omega$, or, equivalently $A_\omega$ and $B_\omega$. We proceed in the same way as in Sect. 6.2:

$$P_{\overline{\mathrm{sp}}\{\mathbf{t^0},\mathbf{t^1},\dots,\mathbf{t^m},\mathbf{Gc_\omega},\mathbf{Gs_\omega}\}}|X\rangle = \sum_{k=0}^{m} \widehat{\gamma}_k|t^k\rangle + \widehat{A}_\omega|c_\omega\rangle + \widehat{B}_\omega|s_\omega\rangle = V_{m+3}|\widehat{\Phi}_\omega\rangle, \tag{122}$$

where

$$V_{m+3} = \begin{pmatrix} | & & | & | & | \\ |t^0\rangle & \dots & |t^m\rangle & |c_\omega\rangle & |s_\omega\rangle \\ | & & | & | & | \end{pmatrix}, \text{ and } |\widehat{\Phi}_\omega\rangle = \begin{pmatrix} \widehat{\gamma}_0 \\ \vdots \\ \widehat{\gamma}_m \\ \widehat{A}_\omega \\ \widehat{B}_\omega \end{pmatrix}. \tag{123}$$

We can write: $P_{\overline{\mathrm{sp}}\{\mathbf{t^0},\mathbf{t^1},\dots,\mathbf{t^m},\mathbf{Gc_\omega},\mathbf{Gs_\omega}\}} = W_{m+3}(W'_{m+3}W_{m+3})^{-1}W'_{m+3}$, where $W_{m+3}$ is identical to $V_{m+3}$ except in the last two columns, where the cosine and sine are tapered by $G$. We thus obtain

$$|\widehat{\Phi}_\omega\rangle = (W'_{m+3}V_{m+3})^{-1}W'_{m+3}|X\rangle, \tag{124}$$

and

$$\widehat{E}_\omega^2 = \widehat{A}_\omega^2 + \widehat{B}_\omega^2 = \widehat{\Phi}_\omega(m+2)^2 + \widehat{\Phi}_\omega(m+3)^2, \tag{125}$$

where $\widehat{\Phi}_\omega(m+2)$ and $\widehat{\Phi}_\omega(m+3)$ are the two last components of vector $|\widehat{\Phi}_\omega\rangle$.

## 6.4 With WOSA

The signal being stationary, we can estimate the amplitude on overlapping segments and take the average. That gives a better estimation, more robust against the background noise, but it has the disadvantage of widening the peaks and thus reducing the resolution in frequency. We simply take Eq. (124), apply it to each segment[7], and compute the average. We have

$$\widehat{E}_\omega^2 = \frac{1}{Q(\omega)} \sum_{q=1}^{Q(\omega)} [\widehat{\Phi}_{q,\omega}(m+2)^2 + \widehat{\Phi}_{q,\omega}(m+3)^2]. \tag{126}$$

## 6.5 Amplitude periodogram or periodogram?

So far, we have studied in detail the periodogram and its confidence levels as well as the estimated amplitude. Of course, confidence levels can also be determined for the amplitude, with Monte-Carlo simulations, or with an

---

[7]We remind that the vectors $|t^k\rangle$ associated to the trend are taken on the whole time series. Only the (tapered) cosine and sine are taken on the WOSA segment.





analytical approximation similar to Sect. 5.3.3.

In the regularly sampled case, at Fourier frequencies, the cosine and sine vectors are orthogonal, so that, in the non-tapered case and with a constant trend, there is no difference between the periodogram and the amplitude periodogram, up to a multiplicative constant. Even with WOSA segmentation, the number of data points being

identical on each segment, that multiplicative constant remains invariant.

In the irregularly sampled case, choosing one or the other depends on what one wants to conserve. The periodogram conserves the flatness of the white noise pseudo-spectrum (see Eq. (87)), while the amplitude periodogram gives a direct access to the estimated signal amplitude. Another criteria to take into account is the computing time. Indeed, the amplitude periodogram requires matrix inversions (or, equivalently, resolution of linear systems) and is then

slower to compute, while the periodogram allows to deal with orthogonal projections and is computationally more efficient. Finally, we mention that, with a trendless signal, difference between both is rather explicit (see Eq. (112)):

Periodogram: $||\widehat{A}_{\omega}|c_{\omega}\rangle + \widehat{B}_{\omega}|s_{\omega}\rangle||^2$ (127)

vs

Amplitude periodogram: $\widehat{A}_{\omega}^2 + \widehat{B}_{\omega}^2$. (128)

This is variance (multiplied by the number of data points) versus squared amplitude. A compromise between the amplitude periodogram and the periodogram is the *weighted periodogram*, which is defined in the next section.

## 7   The weighted WOSA periodogram

Taking into account the approximate linearity between the amplitude periodogram and the tapered periodogram, Eq. (121), a possibility is to perform the spectral analysis with a weighted version of the WOSA periodogram. On each WOSA segment, the periodogram is weighted by $w_q = 2\mathrm{tr}(G_q^2)/\mathrm{tr}(G_q)^2$, $q = 1, ..., Q(\omega)$. The advantage of the weighted WOSA periodogram is to provide deterministic peaks (coming from $A_{\omega}|c_{\omega}\rangle + B_{\omega}|s_{\omega}\rangle$) of more or less equal power on all the WOSA segments, thus alleviating the issue stated in Sect. 4.5.2. The disadvantage is that the

pseudo-spectrum of a white noise is not flat anymore (Eq. (87) is not valid anymore, except when $Q = 1$). Working with the weighted version is done by modifying matrix $M_{2,\omega}$, Eq. (63), which is now

$$M_{2,\omega} = \frac{1}{\sqrt{Q(\omega)}}\left( \begin{array}{ccccc} | & | & & | & | \\ \sqrt{w_1}|h_{1,1}(\omega)\rangle & \sqrt{w_1}|h_{2,1}(\omega)\rangle & \ldots & \sqrt{w_Q(\omega)}|h_{1,Q(\omega)}(\omega)\rangle & \sqrt{w_Q(\omega)}|h_{2,Q(\omega)}(\omega)\rangle \\ | & | & & | & | \end{array} \right).$$ (129)

Note that the weights $w_q$ are the same on each segment when the time series is regularly sampled, so that the whole WOSA periodogram is, in that case, just multiplied by a constant, and the pseudo-spectrum of a white noise is flat.



We observed that the weighted periodogram is often very close to the amplitude periodogram, like in the example presented in Fig. 10. We thus recommend the use of the weighted WOSA periodogram in most spectral analyses. When filtering is to be performed, the amplitude periodogram must be computed as well. This is the topic of the next section.

## 8 Filtering

We want to reconstruct the deterministic periodic part, $\widehat{A}_{\omega}|c_{\omega}\rangle + \widehat{B}_{\omega}|s_{\omega}\rangle$ of our model (12). From Eq. (124), we can extract $\widehat{A}_{\omega} = \widehat{\Phi}_{\omega}(m+2)$ and $\widehat{B}_{\omega} = \widehat{\Phi}_{\omega}(m+3)$, and reconstruction at a single frequency is therefore direct. Reconstruction on a frequency range can be done by summing $\widehat{A}_{\omega}|c_{\omega}\rangle + \widehat{B}_{\omega}|s_{\omega}\rangle$ over $\omega$.

Note that, in theory, reconstruction could be done segment by segment, using the WOSA method. But, in practice,
we observe that it does not give good results with stationary signals. Of course, if the signal is not stationary, reconstruction segment by segment is a clever choice, but, with such signals, it is better to use more appropriate tools such as the wavelet transform. See the second part of this study (Lenoir and Crucifix, 2017), in which some examples of filtering are given.

## 9 Application on paleoceanographic data

The time series we use to illustrate the theoretical results is the benthic foraminiferal $\delta^{18}O$ record from (Jian et al., 2003) that holds 608 data points with distinct ages and covers the last 6 million years. An example of frequency analysis is described below.

### 9.1 Preliminary analysis

We first look at the sampling. $\Delta t_{\text{GCD}} = 1$ kyr, and $r_t = 10.13$ %. Following the recommendation of Sect. 4.4, we
therefore use the default rectangular window taper. The sampling and its distribution are drawn on Fig. 5. We then choose the degree of the polynomial trend to be $m = 7$, see Fig. 6. This choice for $m$ is justified by a sensitivity analysis performed in Sect. 9.4. We remind that the time series is not detrended before estimating the spectral power of the data, but it is detrended before estimating the confidence levels.

### 9.2 CARMA(p,q) background noise analysis

We choose the order of the background noise CARMA process. We opt for the traditional red noise background (Hasselmann, 1976), $p = 1$ and $q = 0$. Note that we observe similar confidence levels with other choices (see the sensitivity analysis in Sect. 9.5). We then estimate the parameters of the stationary CARMA process (here, a red noise) on the detrended data. This is done with the algorithm provided by (Kelly et al., 2014) (see Sect. 5.2.3). Quality of the fit is analyzed on Fig. 7a, 7c and 7e. Fig. 7a analyzes the residuals. If the detrended data are a





Gaussian red noise, the residuals must be distributed as a Gaussian white noise. We see that the distribution is indeed close to a Gaussian. Fig. 7c shows the autocorrelation function (ACF) of the residuals. If the residuals are a Gaussian white noise sequence, they must be uncorrelated at any lag. We can therefore arrange the residuals on a regular grid with a unit step and then take the classical ACF, which can only be applied to regularly sampled data.

Fig. 7c is consistent with the assumption that the residuals are uncorrelated. Fig. 7e shows the ACF of the squared residuals. If the residuals are a Gaussian white noise sequence, the squared residuals are a white noise sequence (which is not Gaussian anymore) and must therefore be uncorrelated at any lag. Deviations from the confidence grey zone indicate that the variance is changing with time and the signal is therefore not stationary. This is actually what is happening with our time series. Changes in variance are already visible on the raw time series (Fig. 6). Remember

that, at this stage, we are within the world of the null hypothesis, Eq. (70), and slight violation of the goodness of fit may be due to the presence of additive periodic deterministic components, that is the alternative hypothesis.

The marginal posterior distributions of the CARMA parameters are shown on Fig. 7b, 7d and 7f, jointly with the ACF of the MCMC samples. Each distribution is unimodal, and we may therefore use the analytical approach of Sect. 5.3.3 to estimate the confidence levels. Based on the ACFs of the MCMC samples of the three parameters, we

skim off the initial joint distribution of the parameters to make their samples almost uncorrelated. In this example, we pick up 1334 samples among the 16000 initial ones. This number of 1334 samples results from the fact that we impose an ACF which is less than 0.2 for each marginal distribution[8].

### 9.3 Frequency analysis

We compute the weighted WOSA periodogram of Sect. 7. The frequency range is automatically determined from

the results of Sect. 4.5.6. The length of the WOSA segments depends on the required frequency resolution. Here we choose segments of about 600 kyr and a 75 % overlapping. The WOSA segmentation is presented on Fig. 8a.

The weighted WOSA periodogram and its 95 % and 99.9 % confidence levels are presented on Fig. 8c and 8d. Both figures display the analytical confidence levels, which are computed with the median parameters of the red noise process (that is the median of 1334 samples of the distributions shown on Fig. 7b, 7d and 7f) and a 12-moment gamma-

polynomial approximation (Sect. 5.3.3). We can check for the convergence of the gamma-polynomial approximation, at some frequencies. This is presented on Fig. 9. Fig. 8c also show the MCMC confidence levels, computed from 50000 red noise time series, all generated with the median red noise parameters. As we can see on Fig. 8c, the matching between the analytical and MCMC confidence levels is excellent, also in the very tail of the distribution, at the 99.9 % confidence level. We can go a step further and take into account the uncertainty on the CARMA

parameters, as explained in Sect. 5.3.2. Fig. 8d presents the MCMC confidence levels that are computed from 50000 red noise time series, generated with stochastic parameters, that are taken from the joint posterior distribution of

---

[8]As explained in Sect. 9.3, these 1334 samples are then used to compute the median parameters, producing the analytical confidence levels of Fig. 8c and 8d and the MCMC confidence levels of Fig. 8c. The MCMC confidence levels of Fig. 8d are computed from 50000 samples of the parameters, after skimming off a distribution with much more samples.



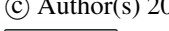

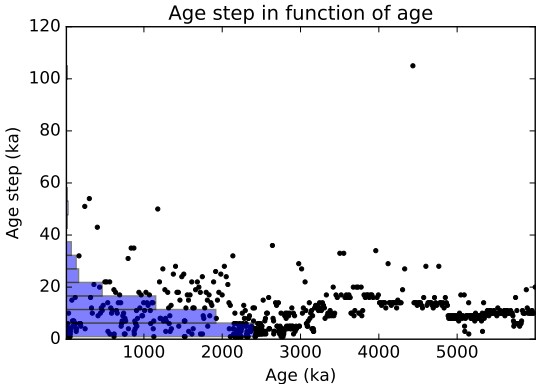

**Figure 5.** The age step, $(t_k - t_{k-1}) \, \forall k \in 2, ..., N$, and its distribution.

the parameters of the red noise process. The number of WOSA segments per frequency, denoted by $Q(f)$ in Sect. 4 to 7, is on Fig. 8b, and provides an indication of the noise damping per frequency. Indeed, the variability due to the background noise is increasingly damped as the number of WOSA segments grows.

We also compute the amplitude periodogram, Eq. (126), which is actually very close to the weighted periodogram,

5    as shown on Fig. 10. Similar results are obtained using other tapers (not shown). This illustrates the quality of the approximations made in Sect. 6.2.2. Note that the estimation of the amplitude $E_\omega$ of the model (12) is always biased by the background noise (we observe on Fig. 10 that the peaks emerge from a baseline which is well above zero).

### 9.4   Sensitivity analysis for the degree of the polynomial trend

We show on Fig. 11 that the degree $m$ of the polynomial trend, taken between 5 and 10, does not influence substantially

10    the WOSA periodogram. Below $m = 5$, the trend no longer fits the data correctly (from a mere visual inspection), while above $m = 10$, spurious oscillations may appear.

Note that we do not apply here the Akaike Information criterion (AIC) (Akaike, 1974). Indeed, defining a stochastic model for the trend and estimating its likelihood is quite tedious in our case, since we work with CARMA stochastic processes. Moreover, at this stage, we do not want to choose yet between the orders of the CARMA process.

### 15   9.5   Sensitivity analysis for the order of the CARMA process

Fig. 12 displays the confidence levels for various orders of the CARMA process: $(p,q) = (0,0)$, $(p,q) = (1,0)$, $(p,q) = (2,0)$ and $(p,q) = (2,1)$. It is clear that the CARMA(0,0) (= white noise) does not capture enough spectral variability to perform significance testing and that using a CARMA(2,0) or a CARMA(2,1) is basically equivalent to using a red noise.





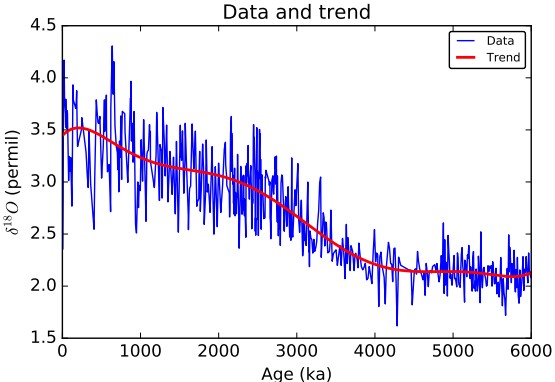

**Figure 6.** The time series and its $7^{th}$ degree polynomial trend.

## 10 WAVEPAL Python package

WAVEPAL is a package, written in Python 2.X, that performs frequency and time-frequency analyses of irregularly sampled time series, significance testing against a stationary Gaussian CARMA(p,q) process, and filtering. Frequency analysis is based on the theory developed in this article, and time-frequency analysis relies on the theory developed in

part II of this study (Lenoir and Crucifix, 2017). It is available at https://github.com/guillaumelenoir/WAVEPAL.

## 11 Conclusions

We proposed a general theory for the detection of the periodicities of irregularly sampled time series. This is based on a general model for the data, which is the sum of a polynomial trend, a periodic component and a Gaussian CARMA stochastic process. In order to perform the frequency analysis, we designed new algebraic operators that

match the structure of our model, as extensions of the Lomb-Scargle periodogram and the WOSA method. A test of significance for the spectral peaks was designed as a hypothesis testing and we investigated in detail the estimation of the percentiles of the distribution of our algebraic operators under the null hypothesis. Finally, we shown that the least squares estimation of the squared amplitude of the periodic component and the periodogram are not longer proportional if the time series is irregularly sampled. Approximate proportionality relations were proposed and are

at the basis of the weighted WOSA periodogram, which is the analysis tool that we recommend for most spectral analyses. The general approach presented in this paper allows an extension to the continuous wavelet transform, which is developed in part II of this study (Lenoir and Crucifix, 2017).

*Code availability.* The Python code generating the figures of this article is available in a supplementary material.







**Figure 7.** CARMA(1,0) background noise analysis. (a), (c) and (e) assess the fit. (a) Standardized residuals. (c) ACF of the residuals. (e) ACF of the squared residuals. The lag refers to an arbitrary scale on which the data are regularly spaced with a unit step. The grey portion is the 95 % confidence region. (b), (d) and (f) show the samples of the MCMC and the posterior marginal distributions, as well as the ACF of the MCMC samples. (b) Mean. (d) Standard deviation of the white noise term. (f) $\log(\alpha)$, where $\alpha$ is defined in Sect. 3.2.3.



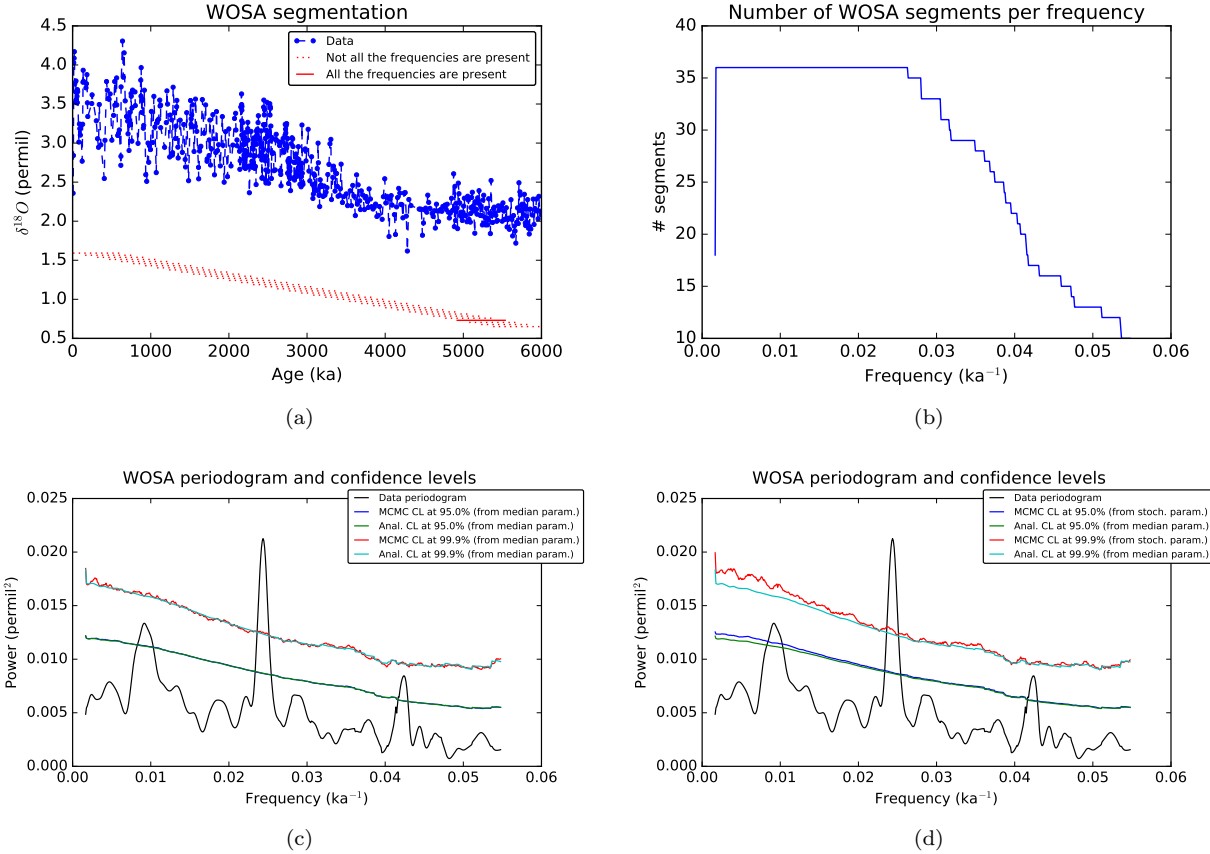

**Figure 8.** Frequency analysis. (a) The time series, in blue, and the WOSA segments, in red. (b) Number of WOSA segments per frequency. (c) and (d): Weighted WOSA periodogram and the confidence levels (CL) at 95 % and 99.9 %. Analytical CL (Anal. CL) are computed with the median parameters of the red noise process. In (c), the MCMC CL are computed from the MCMC red noise time series, all generated with the median red noise parameters. In (d), the MCMC CL are computed from the MCMC red noise time series, generated with stochastic parameters, that are taken from the joint posterior distribution of the parameters of the red noise process.

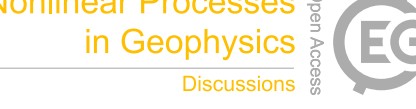


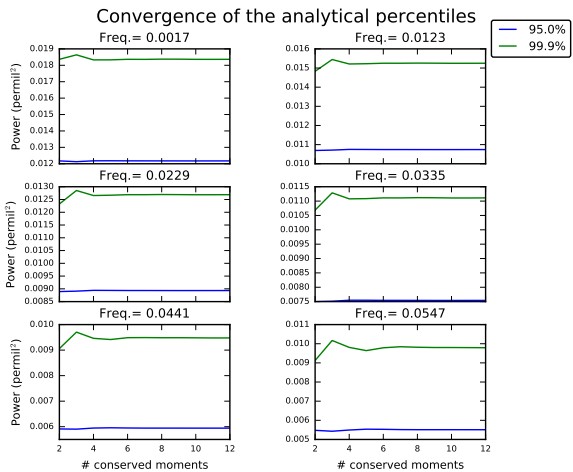

**Figure 9.** At six particular frequencies, check for the convergence of the analytical percentiles.

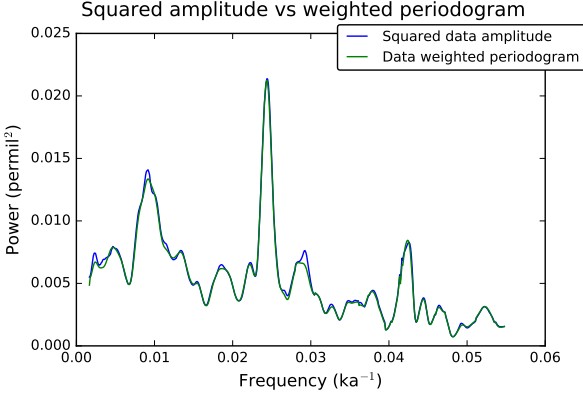

**Figure 10.** Comparison between the amplitude periodogram (= squared amplitude) and the weighted periodogram. The green curve is the same as the black curve of Fig. 8c and 8d

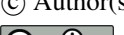



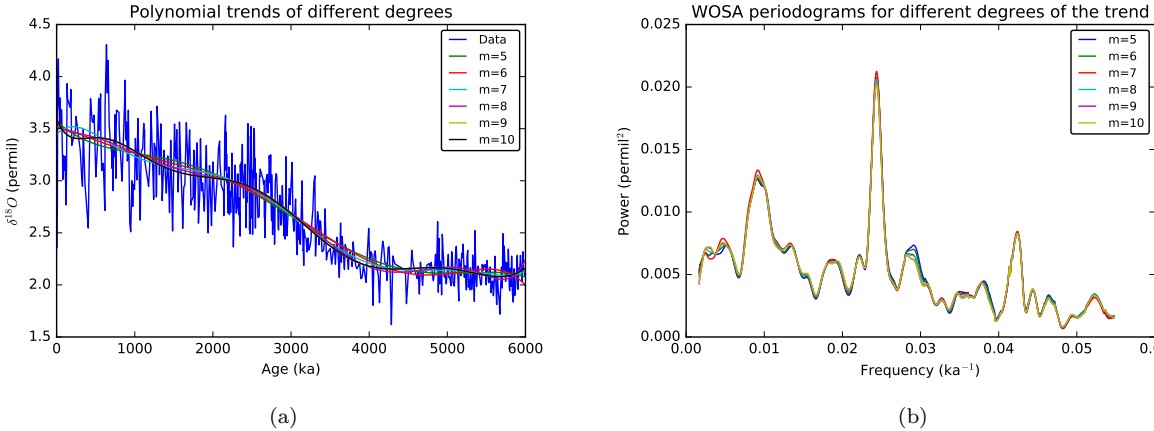

**Figure 11.** (a) Trends of different degrees for the time series. (b) Weighted WOSA periodograms for different degrees of the trend.

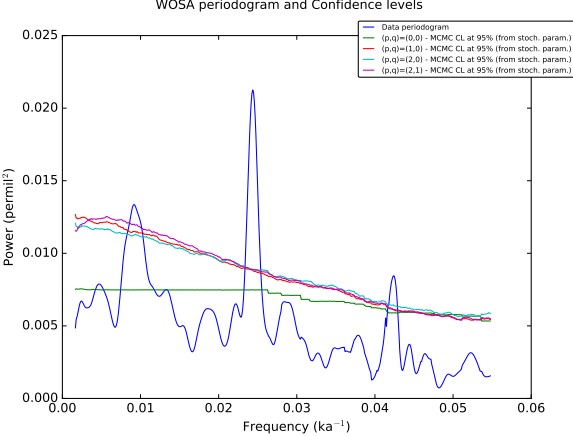

**Figure 12.** The weighted WOSA periodogram and its 95 % confidence levels for different orders (p,q) of the CARMA process. Note that the marginal posterior distributions of some parameters of the CARMA(2,0) and CARMA(2,1) processes are multimodal, so the analytical approach cannot be applied, and MCMC confidence levels must therefore be used.


## Appendix A: Some properties of the Lomb-Scargle periodogram

We present some properties of the LS periodogram, defined in Sect. 4.1.

### A1    Periodicity of the periodogram

The LS periodogram and all its generalizations (e.g. Eq. (58)) exhibit a periodicity similar the DFT of regularly

sampled real processes: The periodogram over the frequency range $]-1/2\Delta t_{\mathrm{GCD}}, 1/2\Delta t_{\mathrm{GCD}}]$ repeats itself periodically. Moreover, the periodogram at frequency $-f$ is equal to the periodogram at frequency $+f$. Consequently, we must work at most on the frequency range $[0, 1/2\Delta t_{\mathrm{GCD}}[$ to avoid aliasing.

### A2    Total reconstruction

Integrating the orthogonal projection $P_{\overline{\mathrm{sp}}\{\mathbf{c}_\omega, \mathbf{s}_\omega\}}$ between frequency 0 and $1/2\Delta t_{\mathrm{GCD}}$ does not give the identity op-

erator. We only have an approximate equality. Using Lomb's approximation, given in Eq. (109), and no extra approximation, some algebra gives

$$\int\limits_{0}^{\pi/\Delta t_{\mathrm{GCD}}} \mathrm{d}\omega\, (|c_\omega^\sharp\rangle\langle c_\omega^\sharp| + |s_\omega^\sharp\rangle\langle s_\omega^\sharp|) \approx \frac{2\pi}{N\Delta t_{\mathrm{GCD}}}\mathbb{I}. \tag{A1}$$

It is interesting to compare it with the integration of complex exponentials, which gives exactly the identity operator:

$$\int\limits_{-\pi/\Delta t_{\mathrm{GCD}}}^{\pi/\Delta t_{\mathrm{GCD}}} \mathrm{d}\omega\, |e_\omega^\sharp\rangle\langle e_\omega^\sharp| = \frac{2\pi}{N\Delta t_{\mathrm{GCD}}}\mathbb{I}, \tag{A2}$$

where $|e_\omega^\sharp\rangle = \frac{1}{\sqrt{N}}\exp(i\omega|t\rangle) = \frac{1}{\sqrt{N}}(|c_\omega\rangle + i|s_\omega\rangle)$. The above formula may be interpreted as a form of Parseval's identity. That property of exact reconstruction is, incidentally, at the basis of the multitaper method (Lenoir, 2017). With that property and the no less interesting mathematical properties of the complex exponentials, it is legitimate to ask why we would not work with the projection on a complex exponential instead of a projection on cosine and sine.

The main disadvantage of working with exponentials is the loss of power in the negative frequencies. Indeed, the trendless model (12) can be rewritten as

$$|X\rangle = E_\omega \frac{\exp(i(\omega|t\rangle + \phi_\omega)) + \exp(-i(\omega|t\rangle + \phi_\omega))}{2} + |\mathrm{Noise}\rangle$$

$$= C_\omega |e_\omega\rangle + D_\omega |e_{-\omega}\rangle + |\mathrm{Noise}\rangle, \tag{A3}$$

where $|e_\omega\rangle = \exp(i\omega|t\rangle)$. In the case of irregularly sampled time series, we no longer have, in general, $\langle e_\omega | e_{-\omega}\rangle = 0$,

so that some power is lost in the negative frequencies when projecting on $\overline{\mathrm{sp}}\{\mathbf{e}_\omega\}$. We could then think about performing the projection on $\overline{\mathrm{sp}}\{\mathbf{e}_\omega, \mathbf{e}_{-\omega}\}$, but this does not lead to the identity operator when integrating from frequency $-1/2\Delta t_{\mathrm{GCD}}$ to $+1/2\Delta t_{\mathrm{GCD}}$.




## A3 Invariance under time translation

As stated in (Scargle, 1982), the LS periodogram is invariant under time translation. $P_{\overline{\mathrm{sp}}\{\mathbf{c}_\omega,\mathbf{s}_\omega\}}$ is of course invariant under such a transformation. The result can be generalized to more evolved projections. Indeed, $[P_{\overline{\mathrm{sp}}\{\mathbf{t^0},\mathbf{t^1},\ldots,\mathbf{t^m},\mathbf{c}_\omega,\mathbf{s}_\omega\}} - P_{\overline{\mathrm{sp}}\{\mathbf{t^0},\mathbf{t^1},\ldots,\mathbf{t^m}\}}]$ is also invariant under time translation, provided all the powers of $|t\rangle$ from 0 to $m$ are taken into account. That projection is also invariant under time dilatation if the frequency is contracted accordingly.

## Appendix B: Periodogram and mean: Equivalence between published formulas

We show here the equivalence between some published formulas, with notations that are a mix between those of the cited articles and those of the present one, in order to facilitate the reading.

In (Brockwell and Davis, 1991, p. 335) the authors work with

$$||(P_{\overline{\mathrm{sp}}\{\mathbf{t^0},\mathbf{c}_\omega,\mathbf{s}_\omega\}} - P_{\overline{\mathrm{sp}}\{\mathbf{t^0}\}})|X\rangle||^2. \tag{B1}$$

It is defined for regularly sampled time series, and is suitable for irregularly sampled time series as well. That formula is the same as Eq. (43).

In (Ferraz-Mello, 1981), the author considers irregularly sampled time series and defines the intensity (p. 620) by

$$I(\omega) = c_1^2 + c_2^2, \tag{B2}$$

where $c_1 = \langle f|h_1\rangle$ and $c_2 = \langle f|h_2\rangle$. $|f\rangle$ contains the measurements (this is $|X\rangle$ in the present article) and $|h_1\rangle$ and $|h_2\rangle$ are exactly the same as in Eq. (47). $I(\omega)$ is thus equal to Eq. (49).

In (Heck et al., 1985), the authors deal with irregularly sampled time series. Equation (1) of (Heck et al., 1985, p. 65) is

$$\mathrm{SP}(\nu) = \langle X|F_{1,0}(\nu)|X\rangle = \langle X|A(\nu)[A(\nu)'A(\nu)]^{-1}A(\nu)'|X\rangle, \tag{B3}$$

where $\nu$ denotes the frequency ($\nu = \omega/2\pi$) and $A(\nu)$ is a $(N,2)$ matrix whose first column is $|c_\omega\rangle - |t^0\rangle\langle t^0|c_\omega\rangle/N$ and second column is $|s_\omega\rangle - |t^0\rangle\langle t^0|s_\omega\rangle/N$. Equation (B3) is nothing but the squared norm of the orthogonal projection of the data $|X\rangle$ onto the span of those two vectors. By a Gram-Schmidt orthonormalization, it is easy to see that $\overline{\mathrm{sp}}\{\mathbf{c}_\omega - \mathbf{t^0}\langle t^0|c_\omega\rangle/N, \mathbf{s}_\omega - \mathbf{t^0}\langle t^0|s_\omega\rangle/N\} = \overline{\mathrm{sp}}\{\mathbf{h_1},\mathbf{h_2}\}$, where $|h_1\rangle$ and $|h_2\rangle$ are defined in Eq. (47). We thus have the periodogram defined in Eq. (49).

## Appendix C: On the pseudo-spectrum

We define the *pseudo-spectrum* as the expected value of the WOSA periodogram under the null hypothesis (see Sect. 5.1):

$$\widehat{S}(\omega) = E\left\{||P_{\mathrm{WOSA}}(\omega)|X\rangle||^2\right\}, \tag{C1}$$



where $|X\rangle = |\text{Trend}\rangle + |\text{Noise}\rangle$, in which $|\text{Noise}\rangle$ is a zero-mean stationary Gaussian CARMA process sampled at the times of $|t\rangle$, and the expectation is taken on the samples of the CARMA noise. With what we have seen in Sect. 5.3.2 and 5.3.3, the periodogram is either obtained with Monte-Carlo methods or analytically with some approximations. In the former case, $\widehat{S}(\omega)$ is estimated by taking the numerical average of the periodogram at each frequency. In the latter case, an analytical formula for the pseudo-spectrum is available. Indeed, the process under the null hypothesis is $|X\rangle = K|Z\rangle + \sum_{k=0}^{m} \gamma_k |t^k\rangle$, where K is defined in Eq. (15) or (34), and we have

$$\widehat{S}(\omega) = \sum_{k=1}^{2Q(\omega)} \lambda_k(\omega) = \text{tr}(M'_{2,\omega} KK' M_{2,\omega}), \tag{C2}$$

where the different terms are defined in theorem 1.

When dealing with a trendless signal, we can perform the WOSA on the classical tapered periodogram and the pseudo-spectrum becomes

$$\widehat{S}(\omega) = E\left\{||P_{\text{WOSA}}(\omega)|X\rangle||^2\right\} = E\left\{\sum_{q=1}^{Q(\omega)} ||P_{\overline{\text{sp}}\{\mathbf{G_q c}_{\omega,\mathbf{q}}, \mathbf{G_q s}_{\omega,\mathbf{q}}\}} |X\rangle||^2\right\}. \tag{C3}$$

In the case of regularly sampled data, Eq. (C3) converges to the *spectrum* $S(\omega)$ as the number of data points increases (up to a multiplicative factor $\Delta t$, the time step). See (Walden, 2000) where it is shown that $||P_{\text{WOSA}}(\omega)|X\rangle||^2$ is a mean-square-consistent and asymptotically unbiased estimator of the spectrum. The *spectrum* $S(\omega)$, also called *Fourier power spectrum*, of a regularly sampled zero-mean real stationary process $|X\rangle$ is defined by (Brockwell and Davis, 1991, Sect. 10.3)[9]:

$$S(\omega) = \Delta t \lim_{N\to\infty} E\left\{||P_{\overline{\text{sp}}\{\mathbf{c}_\omega, \mathbf{s}_\omega\}} |X\rangle||^2\right\}. \tag{C4}$$

Now, considering Eq. (90), we thus have, for trendless regularly sampled time series, the following 1-moment approximation:

$$||P_{\text{WOSA}}(\omega)|X\rangle||^2 \stackrel{d}{\approx} \frac{1}{2Q} S(\omega) \chi^2_{2Q}. \tag{C5}$$

With that approximation, the spectrum $S(\omega)$, which is well known for some processes like ARMA processes, gives access to the confidence levels. The above formula is widely used in the literature on regularly sampled time series in the case of one WOSA segment ($Q = 1$), for which the one moment approximation is good enough, see e.g. (Torrence and Compo, 1998, Eq. (17))).

In the case of irregularly sampled data, $S(\omega)$ can be defined over the frequency range $[-1/2\Delta t_{\text{GCD}}, 1/2\Delta t_{\text{GCD}}[$. This follows from the spectral representation theorem (Priestley, 1981, Chap. 4) applied to irregularly sampled time series. But $\widehat{S}(\omega)$ usually strongly differs from $S(\omega)$. Building estimators of the spectrum $S(\omega)$ in the case of irregularly sampled time series actually seems very challenging, as briefly discussed in Sect. 4.5.1.

---

[9]In that book, the authors work with the projection on complex exponentials, $|e_\omega\rangle = |c_\omega\rangle + i|s_\omega\rangle$, instead of a projection on cosine and sine. But this is asymptotically the same since, asymptotically, the cosine and sine are orthogonal at all the frequencies.



## Appendix D: The generalized gamma-polynomial distribution as an approximation for the linear combination of chi-square distributions

We extend the gamma-polynomial approximation of Sect. 5.3.3 to the *generalized* gamma-polynomial approximation. Both conserve the first $d$ moments of the distribution $X$. The generalized gamma-polynomial approximation is based

on the generalized gamma distribution, which has three parameters, such that the prerequisite of a d-moment approximation is a 3-moment approximation with the generalized gamma distribution.

### D1 3-moment approximation

We work with the generalized gamma distribution, which has 3 parameters,

$$X \overset{d}{\approx} \gamma_{\alpha,\beta,\delta}. \tag{D1}$$

Its PDF is

$$f_\gamma(x;\alpha,\beta,\delta) = \frac{\delta}{\beta^{\alpha\delta}\Gamma(\alpha)} x^{\alpha\delta-1} \exp(-(x/\beta)^\delta) \qquad \alpha,\beta,\delta > 0, \tag{D2}$$

where $\Gamma$ is the gamma function. It reduces to the gamma distribution when $\delta = 1$. Its moments are

$$\mu(k) = \beta^k \frac{\Gamma(\alpha + k/\delta)}{\Gamma(\alpha)} \qquad k \in \mathbb{N}. \tag{D3}$$

Equating the first 3 moments ($k = 1, 2, 3$) of the generalized gamma to the first 3 moments of $X$ gives $\alpha$, $\beta$ and $\delta$.

But, that requires to find the zeros of a nonlinear 3-dimensional function. We observed that root-finding algorithms may be sensitive to the choice of the first guess, and a particular attention must therefore be dedicated to it.

In (Stacy and Mihram, 1965), it is shown that, if $Y$ follows a generalized gamma distribution, working with $\ln(Y)$ allows to find easily the parameters $\alpha$, $\beta$, $\delta$. Indeed, it only requires a root-finding for a monotonic unidimensional function. Unfortunately, the distribution of the logarithm of a linear combination of chi-square distributions is not

known. We thus use the 2-moment approximation, for which we can find the moments of the logarithm of the distribution. Indeed, if we write $Y \overset{d}{=} g\chi_M^2$, in which $g$ and $M$ are determined from Eq. (92), and $Z = \ln(Y)$, some calculus gives us the cumulant generating function of $Z$:

$$K(t) = t\ln(2g) + \ln(\Gamma(M/2 + t)) - \ln(\Gamma(M/2)), \tag{D4}$$

from which we obtain the cumulants. The first three are

$$\kappa(1) = \ln(2g) + \psi_0(M/2), \tag{D5a}$$

$$\kappa(2) = \psi_1(M/2), \tag{D5b}$$

$$\kappa(3) = \psi_2(M/2), \tag{D5c}$$



where $\psi_i$ is the polygamma function ($\psi_0$ is the digamma function). From the cumulants, we have the expected value $\kappa(1)$, the variance $\kappa(2)$, and the skewness $\kappa(3)/\kappa(2)^{3/2}$. Applying (Stacy and Mihram, 1965, Eq. (21)) gives us the parameters $\alpha_0$, $\beta_0$, $\delta_0$ for $Y$, parameters that we then use as a first guess for the generalized-gamma approximation of $X$.

## 5  D2    d-moment approximation

We extend here the formulas[10] presented in (Provost et al., 2009). Let $f_X$ be the PDF of $X$. $f_X$ is approximated by the PDF of a $d^{\text{th}}$ degree generalized gamma-polynomial:

$$f_X(x) \approx \gamma_{\alpha,\beta,\delta}(x) \sum_{i=0}^{d} \xi_i x^i, \qquad x \geq 0, \tag{D6}$$

where the parameters $\alpha$, $\beta$ and $\delta$ are estimated with the above 3-moment approximation. $\xi_0, ..., \xi_d$ are the solution 10 of Eq. (94), where $\eta(h) = \beta^h \Gamma(\alpha + h/\delta)/\Gamma(\alpha)$. The estimation of a confidence level for the WOSA periodogram is then the solution $c_0$ of

$$\frac{1}{\Gamma(\alpha)} \sum_{i=0}^{d} \xi_i \beta^i \gamma(i/\delta + \alpha, (c_0/\beta)^\delta) - p = 0, \tag{D7}$$

for some p-value $p$, e.g. $p = 0.95$ for a 95 % confidence level. If we pose $\delta = 1$, the *generalized gamma-polynomial* approximation reduces to the *gamma-polynomial* approximation presented in Sect. 5.3.3.

## 15  Appendix E: Computing time: Analytical versus Monte-Carlo significance levels

A comparison between the computing times, for generating the WOSA periodogram, with the analytical and with the MCMC significance levels, based on the hypothesis of a red noise background, are presented on Fig. E1. They are expressed in function of the number of data points, which are disposed on a regular time grid, in order to make a meaningful comparison. Confidence levels with the analytical approach are estimated with a 10-moment 20 approximation, and the number of samples for the MCMC approach is 10000 for the $95^{\text{th}}$ percentiles and 100000 for the $99^{\text{th}}$ percentiles. The other parameters are default parameters of WAVEPAL. All the runs were performed on the same computer[11].

We see that the analytical approach is faster than the MCMC approach as long as the number of data points is below some threshold, the latter increasing with the level of confidence. Indeed, the analytical approach delivers 25 computing times of the same order of magnitude whatever is the percentile (the two blue curves in Fig. E1a and E1b are in the same order of magnitude), unlike the MCMC approach, which must require more samples as the level of confidence increases, in order to keep a sufficient accuracy. The difference between both computing times therefore increases as the level of confidence increases.

---

[10]In (Provost et al., 2009), formulas are given for the *gamma*-polynomial, but as suggested by the authors, they can easily be generalized to the *generalized gamma*-polynomial.

[11]CPU type: SandyBridge 2.3 GHz. RAM: 64GB.





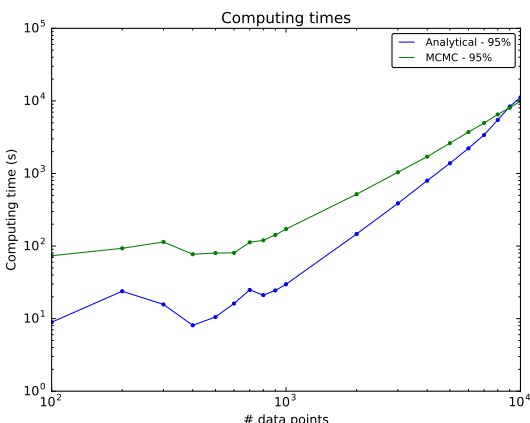 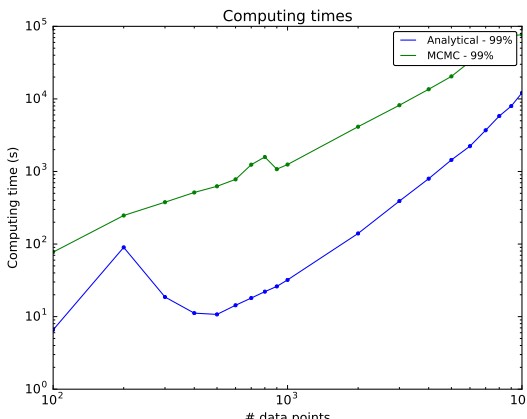

**Figure E1.** Computing times for generating the WOSA periodogram with analytical (blue) and MCMC (green) confidence levels, in function of the number of data points (disposed on a regular time grid). Log-log scale. Left: $95^{\text{th}}$ percentiles. Right: $99^{\text{th}}$ percentiles.

## Appendix F: On the F-periodogram

The formula of the F-periodogram (Eq. (98)) is based on (Brockwell and Davis, 1991, pp. 335-336). In that book, the authors work with a constant trend. We have generalized the formula in order to deal with a polynomial trend. A slightly different formula was published in (Heck et al., 1985, p. 65), again with a constant trend. The F-periodogram is denoted by $\boldsymbol{\theta}_F$ in their paper. In the case of a generalization to a polynomial trend, their formula becomes

$$\frac{(N-2)||(P_{\overline{\text{sp}}\{\mathbf{t^0},\mathbf{t^1},...,\mathbf{t^m},\mathbf{c}_\omega,\mathbf{s}_\omega\}} - P_{\overline{\text{sp}}\{\mathbf{t^0},\mathbf{t^1},...,\mathbf{t^m}\}})|X\rangle||^2}{2||[\mathbb{I}-(P_{\overline{\text{sp}}\{\mathbf{t^0},\mathbf{t^1},...,\mathbf{t^m},\mathbf{c}_\omega,\mathbf{s}_\omega\}} - P_{\overline{\text{sp}}\{\mathbf{t^0},\mathbf{t^1},...,\mathbf{t^m}\}})]|X\rangle||^2},$$ (F1)

but, unlike Eq. (98), it has a denominator which is not invariant with respect to the parameters of the trend.

*Competing interests.* The authors declare that they have no conflict of interest.



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
