# Peer review of "A general theory on frequency and time-frequency analysis of irregularly sampled time series based on projection methods. I. Frequency analysis"

_Nonlinear Processes in Geophysics, 2017_

## Author Comment (AC1) · 4 Jul 2017

We made use of the latex command \paragraph in section 5.3. It was recommended not to use it, and this results in some issues with section numbering in the npg-compiled document. The correct numbering is listed below.

- Section 5.3.4 should be 5.3.3.1

- Section 5.3.5 should be 5.3.3.1.1

[Figure]

- Section 5.3.6 should be 5.3.3.1.2

- Section 5.3.7 should be 5.3.3.1.3

- Section 5.3.8 should be 5.3.3.2

- Section 5.3.9 should be 5.3.3.2.1

- Section 5.3.10 should be 5.3.3.2.2

- Section 5.3.11 should be 5.3.3.2.3

Numbering in section 5.3 will be corrected in the next version of the article.

---

## Referee Comment (RC1) · Anonymous Referee #1 · 9 Sep 2017

Review of "A general theory on frequency and time-frequency analysis or irregularly sampled time series based on projection methods. I. Frequency analysis. " by Lenoir and Crucifix

Recommendation: Minor revisions

The manuscript develops a new method to analyse irregularly sampled time series which is very important for paleoclimate time series. This is a very useful contribution to data analysis. The manuscript is well written, though I needed to get used to the

mathematical notion. I recommend to accept this manuscript after the authors have addressed my below points.

1) How long can the "gaps" in the time series be? How irregular can the time series be? This should be discussed.

2) Eq. (13): The notation of the SDE does not make sense to me. White noise is not differentiable. Wouldn't it make more sense to write the MA part as an memory kernel?

3) Does your significance test also test for trend significance?

4) All references in the manuscript are in the style (Author, year) but in many cases it should be Author (Year). This should be corrected.

5) Page 18, line 7: "This it" should be "This is"

---

## Referee Comment (RC2) · Anonymous Referee #2 · 2 Oct 2017

This paper proposes a new method for frequency analysis of irregularly sampled time series.

The idea of this manuscript is clear and simple. The authors combines some existing results such as LS periodogram and WOSA, but with some important insight and modification.

I think this paper contributes to the practice use of frequency analysis of irregularly sampled time series. Almost all the algorithm is presented in detail. Numerical simula-
tion is presented along with some useful explanation.

To sumarize, I highly evaluate this paper, and suggest it to be accepted for publication.

---

## Author Comment (AC2) · 4 Dec 2017

We are very grateful to the reviewer for the constructive comments and suggestions. We provide below a point-by-point reply to the reviewer's comments. In addition, during the review process, we got comments from other people that we judged pertinent to include in this revised version. The related minor changes are listed afterwards.
When we mention a section or an equation, we refer to the new version of the manuscript.

[Figure]

**1 How long can the "gaps" in the time series be? How irregular can the time series be? This should be discussed.**

The gaps can a priori be of any length. Indeed, the algorithms are constrained to avoid as much as possible the artifacts caused by aliasing, which are themselves caused by the gaps. This is explained in Sect. 4.5.4 of paper I and Sect. 3.8 of paper II. In paper II, we show that the algorithms do a good job at rejecting areas where artifacts are caused by big gaps, which is probably the most problematic case. The criteria presented in these sections give a basis which could be improved in subsequent studies.

**2 Eq. (13): The notation of the SDE does not make sense to me. White noise is not differentiable. Wouldn't it make sense to write the MA part as a memory kernel?**

This is now Eq. (14). We agree that the white noise is, strictly speaking, not differentiable. Things are now better explained: we still keep the same notation in Eq. (14), like in classical references in the field, e.g. Jones and Ackerson (1990) or Brockwell (2016, Sect. 11.5), but we explain that is interpreted through an Itô differential equation, which makes sense. We also define the white noise from the Brownian motion, in order to keep things clear for the specialists. All of this new material comes from the classical reference of Brockwell and Davis (2016, Sect. 11.5). Here is the modified part of the manuscript:
A CARMA(p,q) process is simply the extension of an ARMA(p,q) process to a continuous time[1]. A zero-mean CARMA(p,q) process $y(t)$ is the solution of the following
* * *
[1]A CARMA(p,q) process sampled at the times of an infinite regularly sampled time series is an ARMA(p,q) process.

stochastic differential equation:

$$\frac{d^p y(t)}{dt^p} + \alpha_{p-1}\frac{d^{p-1}y(t)}{dt^{p-1}} + ... + \alpha_0 y(t) = \beta_q\frac{d^q\epsilon(t)}{dt^q} + \beta_{q-1}\frac{d^{q-1}\epsilon(t)}{dt^{q-1}} + ... + \epsilon(t), \quad (1)$$

where $\epsilon(t)$ is a continuous-time white noise process with zero mean and variance $\sigma^2$. It is defined from the standard Brownian motion $B(t)$ through the following formula:

$$\sigma dB(t) = \epsilon(t)dt \quad (2)$$

The parameters $\alpha_0$, ... , $\alpha_{p-1}$ are the autoregressive coefficients, and the parameters $\beta_1$, ..., $\beta_q$ are the moving average coefficients. $\alpha_p = \beta_0 = 1$ by definition. When $p > 0$, the process is stationary only if $q < p$ and the roots $r_1, ..., r_p$ of

$$\sum_{k=0}^{p} \alpha_k z^k = 0, \quad (3)$$

have negative real parts. Strictly speaking, the derivatives of the Brownian motion $\frac{d^k B}{dt}$, $k > 0$, do not exist, and we therefore interpret Eq. (1) as being equivalent to the following measurement and state equations

$$y(t) = bw(t), \quad (4)$$

and

$$d|w(t)\rangle = A|w(t)\rangle dt + dB(t)|e\rangle, \quad (5)$$

where $|b\rangle = [\beta_0, \beta_1, ..., \beta_q, 0, ..., 0]'$ is a vector of length $p$, $|e\rangle = [0, 0, ..., 0, \sigma]'$, and

$$A = \begin{pmatrix} 0 & 1 & 0 & \ldots & 0 \\ 0 & 0 & 1 & \ldots & 0 \\ \vdots & \vdots & \vdots & \ddots & \vdots \\ 0 & 0 & 0 & \ldots & 1 \\ -\alpha_0 & -\alpha_1 & -\alpha_2 & \ldots & -\alpha_{p-1} \end{pmatrix}. \quad (6)$$

Equation (5) is nothing else but an Itô differential equation for the state vector $|w(t)\rangle$.

**3 Does your significance test also test for trend significance?**

There is no significance test for the trend. We rather build periodograms (Eq. (58), (59) or (64)) that are blind to the parameters of the trend (i.e. they are invariant with respect to these parameters).

**4 All references in the manuscript are in the style (Author, year) but in many cases it should be Author (year). This should be corrected.**

Corrected.

**5 Page 18, line 7: "This it" should be "This is"**

Corrected.

**Other changes**

- Notations: All is bra-ket now, instead of a mix between bra-ket and bold symbols. For example, $\overline{\mathrm{sp}}\{\mathbf{a}\}$ is changed to $\overline{\mathrm{sp}}\{|a\rangle\}$.

- The angular frequency in the model for the data is now denoted $\Omega$ (instead of $\omega$), in order to make the difference between this frequency and the probed frequency by the periodogram which is denoted $\omega$. See Sect. 3.1.

- In Fig. 11b, the periodograms are now normalized according to Eq. (58), which is more rigorous when comparing them. Visually, the results with the

ODP1148 data set are very similar as without this normalization, so that the discussion/interpretation remains unchanged.

**References**

Brockwell, P.J. and Davis, R.A.: Introduction to Time Series and Forecasting, Springer Texts in Statistics, Springer International Publishing, ISBN: 978-3-319-29854-2, Third edn, doi: 10.1007/978-3-319-29854-2, 2016.

Jones, R.H. and Ackerson, L.M.: Serial correlation in unequally spaced longitudinal data, Biometrika, 77, 721-731, doi: 10.1093/biomet/77.4.721, http://biomet.oxfordjournals.org/content/77/4/721.abstract, 1990.

---

## Author Comment (AC3) · 4 Dec 2017

We are very grateful to the reviewer for the comments. During the review process, we got comments from other people that we judged pertinent to include in this revised version. The related minor changes are listed in the reply AC2.

---

## Author Response (AR1)

**A general theory on frequency and time-frequency analysis of irregularly sampled time series based on projection methods. I. Frequency analysis**

Guillaume Lenoir[1] and Michel Crucifix[1,2]

[1]Georges Lemaître Centre for Earth and Climate Research, Earth and Life Institute, Université catholique de Louvain, BE-1348, Louvain-la-Neuve, Belgium
[2]Belgian National Fund of Scientific Research, rue d'Egmont, 5, BE-1000 Brussels, Belgium

*Correspondence to:* Guillaume Lenoir (guillaume.lenoir@hotmail.com)

**This is the manuscript with the tracked modifications. See the interactive discussion for the replies to the reviewers.**

[revised manuscript text omitted]